# Hydraulic Modeling and Evaluation Equations for the Incipient Motion of Sandbags for Levee Breach Closure Operations

Ahmed M. A. Sattar [1,2], Hossein Bonakdari [3,*], Bahram Gharabaghi [3,*] and Artur Radecki-Pawlik [4]

1   Irrigation and Hydraulics Department, Faculty of Engineering, Cairo University, Giza 12613, Egypt; ahmoudy77@yahoo.com
2   Civil department, German University in Cairo, New Cairo City 13611, Egypt
3   School of Engineering, University of Guelph, Guelph, ON N1G 2W1, Canada
4   Institute of Structural Mechanics, Faculty of Civil Engineering, Cracow University of Technology, 31-155 Krakow, Poland; rmradeck@cyf-kr.edu.pl
*   Correspondence: hbonakda@uoguelph.ca (H.B.); bgharaba@uoguelph.ca (B.G.)

**Abstract:** Open channel levees are used extensively in hydraulic and environmental engineering applications to protect the surrounding area from inundation. However, levees may fail to produce an unsteady flow that is inherently three dimensional. Such a failure may lead to a destructive change in morphology of the river channel and valley. To avoid such a situation arising, hydraulic laboratory modeling was performed on an open channel levee breach model capturing velocity, in x, y and z plans, at selected locations in the breach. Sandbags of various shapes and sizes are tested for incipient motion by the breach flow. We found that a prism sandbag has a better hydrodynamic characteristic and more stability than spherical bags with the same weight. Experimental results are then used to evaluate existing empirical equations and to develop more accurate equations for predicting critical flow velocity at the initial stage of sandbag motion. Results showed the superior predictions a few of the equations could be considered with an uncertainty range of ±10%. These equations explained the initial failed attempts of the United States Army Corps of Engineers (USACE) for breach closure of the case study, and confirmed the experimental results are simulating the case study of breach closure.

**Keywords:** breach closure; levee breach; channel side flow; sandbags; open channel flow; critical velocity; incipient motion

## 1. Introduction

Over seven million people have lost their lives in floods between 1990 and 2017 in Asia alone [1,2]. Predicting flood water levels, the design of stable alluvial channels, and protecting lives and properties during major flood events is one of the greatest challenges of the 21st century [3,4]. Open channel levees are used extensively in hydraulic and environmental engineering applications. They are constructed for the purpose of water constriction within the channel and act as an efficient structural method of floodwater control. However, breaches are very likely to occur in levees during high flood events and in locations where levees are not constructed properly. These breaches release flood waves that inundate the surrounding area, causing property damage, social interruptions and can threaten human lives. A severe example is the 2005 levee breach in New Orleans, LA [5]. In an open-channel levee breach, flow is diverted laterally in a hydraulic manner similar to side weir flow and dividing flow [6,7]. The flow in the main channel approaching the breach starts to divide as the flow enters the breach. A separation zone develops in the main channel after the breach, and a stagnation zone is formed near

the downstream corner of the breach. At the breach outflow, a contracting flow region is observed with free over-fall conditions, similar to dam-break flow.

For side weir flow, most of the earlier studies were devoted to the evaluation of the discharge coefficient for various weir geometries, with few studies discussing hydraulic behaviors of flow at the weir exit [8–17]. However, for dividing flows in the open channel, more attention was given by earlier studies to the hydraulic characteristics of dividing flow (e.g., [18–25]). Research in levee breach hydraulics has fallen into two categories: (1) Studying the impact of breach flow on the hydraulics of the main upstream channel, and (2) studying the impact of breach flood waves propagation on downstream urban areas.

Within the first category, Yen [26] studied the impact of levee breach on water levels in the main channel upstream of the breach, and discussed various methods for estimating the channel capacity. Jaffe and Sanders [27] studied engineered levee breaches using a shallow water model for an optimal design to reduce flood stages. Apel et al. [28] investigated the effect of levee breaches on the flood frequency distribution in the main channel. In the second category, Sattar [29] used a 1:50 scale model to investigate flood waves generated by the 17th Street Canal breach in New Orleans following Hurricane Katrina. Using the same scale model, Sylvie et al. [30] compared measurements of urban flooding in the breach neighborhood with the 2D numerical model. LaRocque et al. [31] studied the flow pattern of the New Orleans flooded the neighborhood. The flow characteristics of a side channel breach were studied by Sattar [32]. Others have investigated the interaction of general flood waves with city-like blocks, either experimentally [33,34] or numerically [35–39].

Mitigating the adverse effects of breach floods on urban areas can be achieved using inundation plans, based on which evacuation plans will be devised, or by breach closure and stopping flood wave releases as early as possible. The most efficient way for closure of levee breaches is the utilization of large sand bags. These bags are dropped in flowing water to decrease flooding and eventually close the breach. Following the extreme events of Hurricane Katrina, levees protecting the city of New Orleans breached at several locations, with the 17th Street Canal levee breach being the biggest, threatening the whole Metropolitan Orleans East Bank. The United States Army Corps of Engineers [5] tried to drop large sandbags to close the breach and stop flood waves from inundating the east bank, but the sandbags were washed away with the breach strong flow. This failure to close the breach with sandbags led to the inundation of the entire Orleans east basin.

It is very important to know the hydrodynamic characteristics of the sand bags, especially critical velocity at incipient motion. Although many researchers have studied the subject of particle incipient motion for more than two centuries after Brahms in 1753, the effect of particle shape has not been examined, except in very few studies. The effect of particle shape on the threshold of motion has been investigated using particles with different shapes and sizes and constant density under subcritical, uniform flow conditions in a titling flume [40,41]. The effect of particle shape on the threshold of motion using constant specific weight for all shapes was studied by Gogus and Defne [41].

Gulcu [42] studied the effect of shape and size of individual particles on the initiation of motion over a smooth sloping channel bed when the particles are resting behind an obstruction of known height. In spite of such importance of sandbags for breach closures, relevant studies are still lacking, and hydrodynamic characteristics of sandbags in open channel flow are not reported in the literature [43]. An equation was proposed by Izbash [44] for incipient motion of rocks deposited in running water for the purpose of cofferdam construction. The stability of sandbags placed on non-uniform slopes has been studied by Kobayashi and Jacobs [45]. Additionally, Zhu et al. [43] investigated the hydrodynamic characteristics of sand-filled geosynthetic bags used for the construction of submerged dikes in rivers.

Neill et al. [46] conducted experiments using sandbags to protect the bank from erosion, using a previously developed formulation [5]. In Korkut et al. [47] study, Geobag were used to check the stability of the sandbags. To study the protection of the bridge abutment using sandbags, lab experiments were conducted [47], and the size of the bags and ripraps were determined based on methods in Pilarczyk [48] and Richardson and Davis [49]. El-Kholy and Chaudhry [50] studied the

path of sandbags as they are washed away by flood waves and incipient motion for spherical bags. Juez et al. [51] carried out an extensive study on numerical, hydrodynamic, and morphological models of dam break flow over mobile beds. Their study showed that the interactions between dam break flow and mobile bed have a great impact on obtained results.

The experimental hydraulic laboratory study in this paper is intended to provide more understanding for the hydraulics of breach flows and incipient motion of sandbags used for closure. This study has four main objectives: (1) To provide 3D velocity measurements at selected locations in an open channel levee breach, (2) to investigate the effect of shape and size of sandbags on the threshold of motion, (3) to calibrate existing formulae and develop new ones for incipient motion of sandbags used for breach closure, and finally, (4) to apply developed equations to the case of the 17th Street Canal levee breach closure.

The aim of the study is to perform laboratory modelling for evaluation of the empirical equations for the critical velocity of incipient motion of sandbags for a levee breach closure. It is especially crucial in urbanized areas: In such places, unexpected fluvial processes, against which we use river engineering methods, destroy urban systems that are difficult to re-establish.

The results of this research may be used for further investigations to develop universal optimum open channel breach closure procedures for breaches in other locations and with both erodible and intact levees. In this paper, first a description of the experimental setup used to simulate an open channel levee breach is presented. Sandbags with various shapes and sizes are prepared and tested to measure their velocity for incipient motion under the impact of levee breach waves. The hydrodynamic characteristics of sandbags are discussed with relation to flow hydraulics. Based on experimental hydraulics measurements, two equations for sandbag incipient motions are calibrated, and new equations are developed. Statistical analysis is performed on the developed equations for error and uncertainty bands. Finally, the developed equations are applied to the 17th Street Canal levee breach.

## 2. Equations for Sandbag Incipient Motion

The size of the sandbags can be determined by relating the critical flow conditions at the onset of bag instability to the bag size that is not washed away. The sandbags could then be sized so that they settle at the channel bed to close the breach. Stability conditions can be evaluated using either the critical bed shear stress or the critical velocity at the initiation of sandbag motion. The latter approach is followed herein. The earliest research was conducted by Brahms in 1753, and resulted in the following equation for spherical particles:

$$V_{cr} = C_1 W^{1/6} \tag{1}$$

where $V_{cr}$ = depth averaged velocity at the location of the particle at which the particle starts moving, $W$ = weight of the particle, and $C_1$ = an empirical constant.

Using the particle diameter instead of the weight, Yu et al. [37] suggested the following power function:

$$V_{cr} = 2.5 D^{0.44} \tag{2}$$

The following expression for spherical rocks used in toe dumping in flowing water [42]:

$$V_{cr} = C_2 \sqrt{2gD \left( \frac{\rho_s - \rho_w}{\rho_w} \right)} \tag{3}$$

where $D$ = diameter of the spherical particle, $g$ = gravitational constant, $\rho_s$ = particle density, $\rho_w$ = water density, and $C_2$ = an empirical constant. Novak and Nalluri [52] proposed an equation for the critical motion of single particles on smooth and rough beds in the form

$$V_{cr} = C_3 \sqrt{gD \left( \frac{\rho_s - \rho_w}{\rho_w} \right)} \left( \frac{D}{R} \right)^{C_4} \tag{4}$$

where $R$ is the channel hydraulic radius and was included in the empirical equation to generalize their approach, which has been done using circular shape channels; and $C_3$ and $C_4$ are empirical constants. This relation is stated to be valid for $D/R < 0.3$. On the other hand, for prism sandbags, Gulcu [42] used force balance on single solid particles and the proposed relation between critical velocity and particle dimensions as follow:

$$\frac{V_{cr}}{\sqrt{gb\left(\frac{\rho_s - \rho_w}{\rho_w}\right)}} = \sqrt{\mp C_5 \left(\frac{b}{c}\right)^2 + C_6 \frac{b}{c} + C_7} \tag{5}$$

where $b$ is sandbag height, $c$ is sandbag length parallel to the flow direction, and $C$ represents empirical constants. Zhu et al. [43] worked on sandbags and assumed that the sandbag would start to slide at the incipient condition and to apply a simple force balance equation to maintain static equilibrium, they proposed the following equation for prism sandbags:

$$V_{cr} = C_8 \left(\frac{H}{b}\right)^{1/6} \sqrt{\frac{c}{b}} \sqrt{gb\left(\frac{\rho_s - \rho_w}{\rho_w}\right)} \tag{6}$$

where $H$ is water depth above the sandbag, and $C_8$ is a constant to be calibrated from experimental data.

## 3. Experimental Setup

To model a general case of levee breach failure, experiments were performed in a straight flume, which is constructed in $11.6 \times 6.7$ m basin (Figure 1) at the Department of Irrigation & Hydraulics, Cairo University, Egypt. The experiments were designed to simulate breach flow after full formation of the breach and did not account for breach formation. The flume was 6.4 m long, 0.6 m deep and 0.4 m wide. The channel bed and sides were covered with rough concrete with an average Manning roughness coefficient of 0.01. All experiments were performed on fixed beds, in the channel, and in breach. The discharge was measured by the electromagnetic flow meter installed on the discharge pipe.

The current study was on a single lateral breach located 3 m from the channel entrance to have developed flow in the channel upstream from the breach. The floor of the flume was horizontal, and the breach discharged on a horizontal platform constructed at the same level as the channel. Breach flow was allowed to flow freely without any impact of this boundary on breach flow. The downstream end of the flume was left open with an adjustable tailgate for controlling the water level and discharge in the downstream channel. Two 0.16 m$^3$/s (2500 gallons per minutes) axial pumps were used for flow supply from an underground sump to an overhead tank that supplies the flume through a 0.305 m supply pipe. To ensure uniformly distributed flow at the breach location, perforated screens and a 60 mm long honeycomb were placed at the flume inlet.

A point gauge was used to measure water surface in the channel with an accuracy of 0.1 mm on the Vernier scale. A 10 MHz 3D Sontek Acoustic Doppler Velocimeter (ADV) (SonTek, San Diego, CA, USA) was used to measure 3D velocity and turbulent components. To verify the repeatability of experiments, the tests were conducted several times, and the velocity components at a selected location were recorded. It was found that experiments are repeatable and the difference in velocity was less than 1%.

A coordinate system was chosen such that the positive $x$-axis was in the flow direction of the main channel. The positive $y$-axis was pointing toward the channel side, while the breach and the positive $z$-axis was upward in the vertical direction, as shown in Figure 1a. The origin of this coordinate system was at the bed of the flume at the upstream end of the breach opening. The experiments were performed with the flow rate and the depth downstream of the flume held constant. Measurements were taken after conditions became steady in the channel and flow was fully turbulent for all tests. This experimental setup has been used previously to perform hydraulic analyses for channel levee breach flow [29]. The authors of that study used the grid shown in Figure 1b to measure flow velocity at various locations in the channel and through the breach.

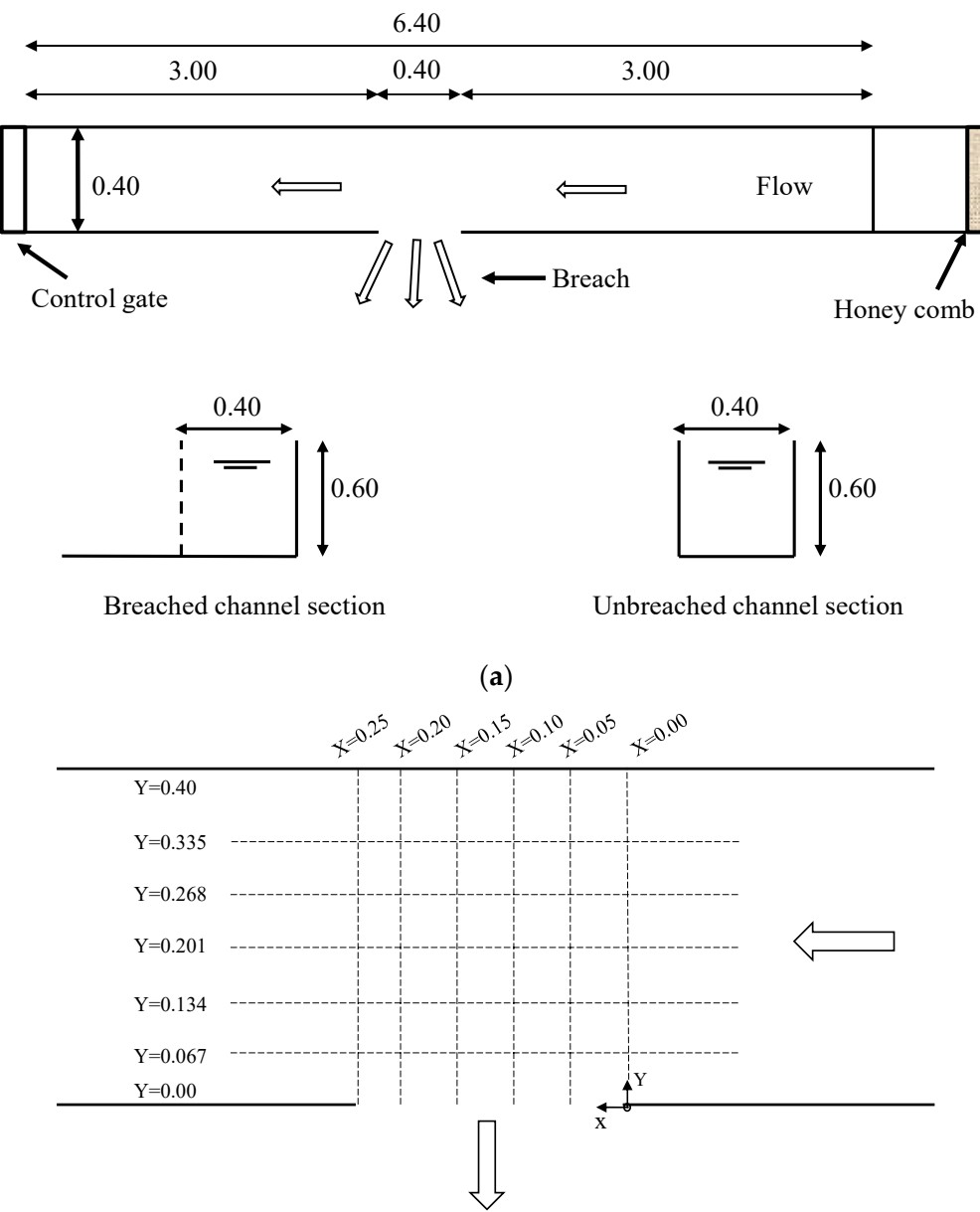

**Figure 1.** Schematic description of the experimental setup. (**a**) The experimental levee breach model, (**b**) the grid used for velocity measurements.

Sandbags of different shapes and sizes were made of sewed cloth bags filled with sand. Densities of material were determined by weighing the bags on a digital balance of 0.01 g accuracy and by measuring their dimensions to calculate the volume. The dimensions of each sandbag were measured using a compass with an accuracy of 0.1 cm. The sandbags had an average density of 1.85 g/cm$^3$. Rectangular prisms were divided into two groups according to their orientation to the flow direction. Geometrical properties and dimensional details of the sandbags used in this investigation are shown in Table 1. It should be noted that *a*, *b*, and *c* are the width, height and length of an imaginary rectangular prism enfolding sandbag, and $\forall$ and S are the volume and surface area, respectively. All sandbags were placed such that the shortest dimension was always the height of the sandbag.

**Table 1.** Geometric and dimensional characteristics of sandbags used in this study.

| Sandbag Shape | Statistical Parameter | $a$ (cm) | $b$ (cm) | $c$ (cm) | $\forall$ (cm$^3$) | S (cm$^2$) |
|---|---|---|---|---|---|---|
| Prism (29 cases) | Minimum | 3 | 2.5 | 4 | 40 | 72 |
| | Maximum | 18 | 7 | 18 | 378 | 384 |
| | Average. | 6.9 | 3.82 | 9.1 | 214 | 236 |
| | Standard deviation | 4.2 | 1.15 | 4.3 | 103 | 93 |
| Sphere (13 cases) | Minimum | | 5 | | 58 | 72 |
| | Maximum | | 22 | | 5575 | 1521 |
| | Average. | | 13 | | 1739 | 618 |
| | Standard deviation | | 6 | | 1788 | 475 |

## 4. Experimental Procedure

Two breach widths $L_b$ were considered (0.25 m and 0.40 m), and for each the downstream channel gate was altered to change flow through the breach and thus the breach flow ratio, $Q_r = Q_b/Q_u$, was changed from 1 to 0.80, where $Q_b$ is the breach discharge and $Q_u$ is the main channel discharge upstream of the breach. $Fr_u$ is the main channel Froude number upstream of the breach, and $Fr_d$ is the main channel Froude number downstream of the breach. Table 2 presents the hydraulic parameters which were recorded in the experimental study.

**Table 2.** Breach hydraulic parameters.

| Breach Width | $Q_u$ (m$^3$/s) | $Q_b$ (m$^3$/s) | $Q_r$ | $Fr_u$ | $Fr_d$ |
|---|---|---|---|---|---|
| 0.25 m | 0.16 | 0.16 | 1 | 0.62 | - |
| | 0.16 | 0.128 | 0.8 | 0.62 | 0.11 |
| 0.4 m | 0.16 | 0.16 | 1 | 0.62 | - |
| | 0.16 | 0.128 | 0.8 | 0.62 | 0.11 |

In each experiment, discharge was kept constant until steady flow conditions had developed in the channel upstream of the breach section. A sandbag with a selected dimension was then placed at the bottom of the flume at the mid-breach section. Sattar [32] extensively studied various procedures for closure of levee breaches and found that staring by dumping sandbags in mid-breach sections is the most effective way for closure. The flow was increased in small increments, and this procedure was repeated until the motion of sandbag was observed. Then the flow was decreased in three steps, and the test was repeated to make sure that it reflected real critical conditions for the incipient motion of the sandbag. This procedure for determining the critical conditions at the incipient motion is defined in detail by Gogus and Defne [41] and Gulcu [42].

Once the criterion for sandbag motion was satisfied, the depth of water above the sandbag was measured; the vertical velocity profile over the flow depth was recorded with ADV. Due to the non-uniform distribution of vertical velocity along the breach section (velocity increases along the breach to reach its maximum at the end of the contraction zone), and the presence of velocity components in both x- and y-directions, the critical velocity for incipient motion was taken as the velocity magnitude averaged over the height of the sandbag at the test location.

## 5. Results and Discussions

This section first presents some aspects of breach hydraulics, including water surface profiles and flow velocity in the x-, y- and z-directions around the breach section. This is followed by presenting measurements of critical velocity for various shapes and sizes of sandbag. These measurements are then used to calibrate exiting equations and develop new equations that are tested statistically and then applied to the 17th Street Canal levee breach in New Orleans.

### 5.1. Water Surface Mapping

The flow in the main channel approaching the breach started to divide as the flow enters the breach. A separation zone developed in the main channel after the breach, and a stagnation zone formed near the downstream corner of the breach. At the breach outflow, a contracting flow region is observed with free overfall conditions similar to dam-break flow. For $L_b$ = 40 cm, and $Q_r$ = 1, water depth in the main channel upstream of the breach was 38 cm, which increased longitudinally across the main channel to 42 cm downstream of the breach, and decreased rapidly in the transverse direction at the breach location to 29 cm at the breach. Figure 2 shows the mapping of water depth for $L_b$ = 40 cm.

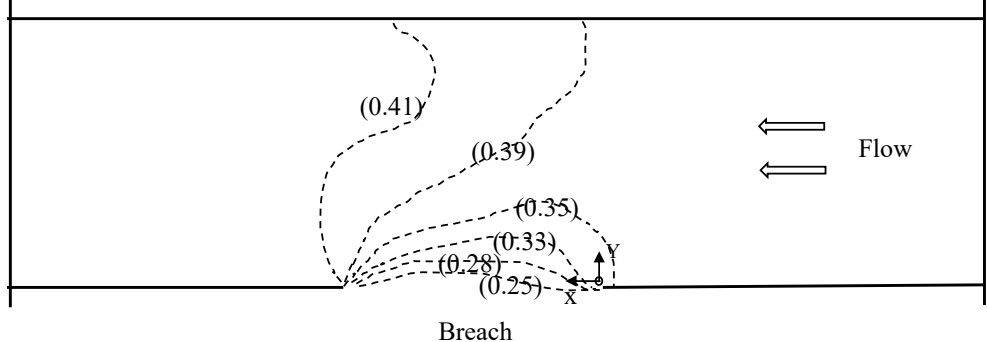

Breach

**Figure 2.** Water depth along the channel for $L_b$ = 40 cm, and $Q_r$ = 1 m$^3$/s.

At the breach section, the variations in water depth were more pronounced than in the case of open channel dividing flow. There was a rapid drop in the water surface from the outer channel bank to the breach location and a corresponding rapid increase in the flow velocity, which was highest at the breach location. Flow depth in the stagnation zone was the highest along the breach section. Due to flow circulation and a significant decrease in the flow velocity in the separation zone, the water depth in this zone was higher than the main channel and was equal to that of the stagnation zone, downstream of the breach.

The highest water depth opposite to the breach line occurred in the separation zone. At $Q_r$ = 1 m$^3$/s, the recirculation effects and the separation zone were highest when compared to those in the other case of $Q_r$ < 1. Unlike for open channel dividing flow, the breach flow separation occurred at the near side of the breach in the main channel, causing a flow contraction. This is obvious from the velocity vectors across the breach. However, for the same $Q_r$, the larger the breach opening, the larger the angle of entry of flow, and the extent of flow separation in the main channel just before the flow exits from the breach. Also, the larger the breach opening, the smaller the contraction zone. Michelazzo et al. [7,53] discussed more details on the influence of breach width on channel flow. These characteristics are similar to those of the dividing open channel flows [20]. Figure 3 shows the hydraulics of an open channel levee breach for the two considered cases, showing clearly the separation and stagnation zones.

### 5.2. Velocity Fields

The following section presents the findings related to velocity measurements in the open channel and side breach. Michelazzo et al. [7,53] have presented 3D velocity measurements for open channel side flow similar to those discussed herein. Figure 4 shows the flow velocity in the x- and y-direction at the breach section in the main channel at z = 0.04 m and z = 0.11 m for $Q_r$ = 1 m$^3$/s. The x-velocity decreases at the mid–section of the breach and reaches zero by the end of the breach due to the complete curvature of the vectors exiting the breach. Experimental results showed that the vectors at the tip of the stagnation zone at the breach end took a steep curvilinear path to exit the breach and were forced by the flume side to exit in the negative x-direction.

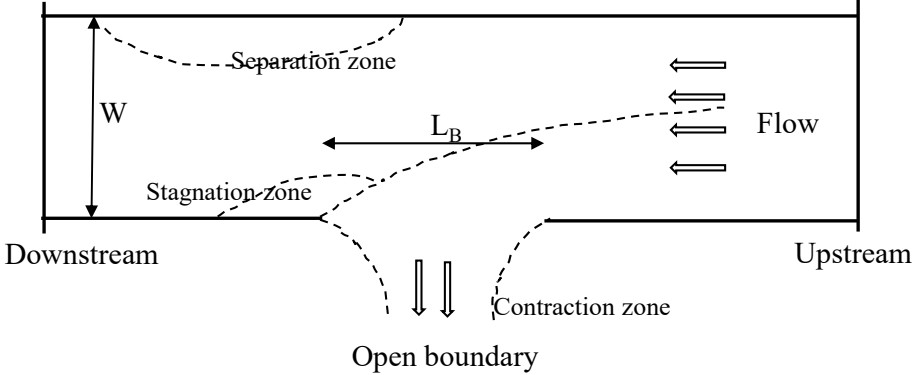

**Figure 3.** Schematics of the hydraulics of channel levee breach (L$_B$ and W are breach width and channel width respectively).

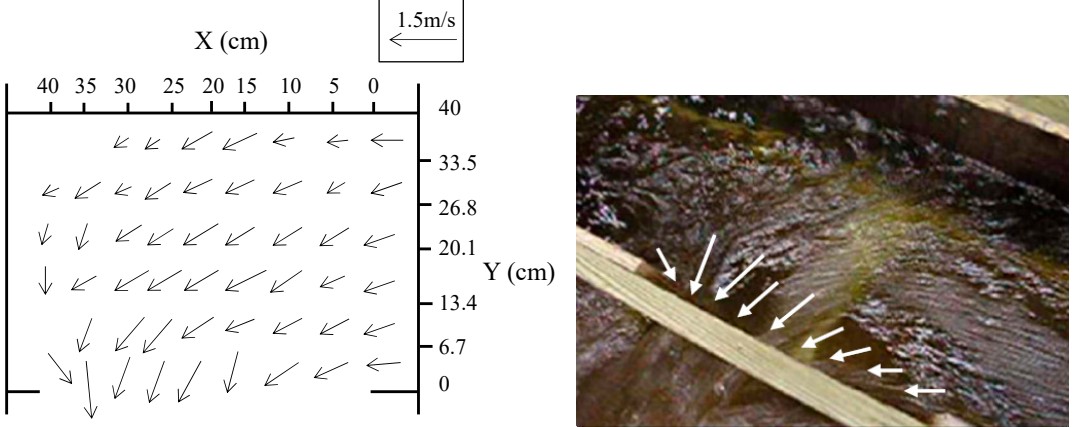

**Figure 4.** x- and y-velocity field at the breach section in the main flume. $Q_u$ = 0.16 m$^3$/s, $Q_r$ = 1 m$^3$/s, z = 0.15 m, for $L_b$ = 0.40 m.

The increase in x-velocity can be clearly observed with the increase in distance from the flume bottom. It can also be observed that the distribution of x-velocity in the main channel around the breach is almost the same at different distances from the bottom, where the changes is only in the values. At the far end of the breach in the main channel x-velocity decreased significantly, and the direction changed at some locations at z = 0.04 m.

The velocity in the y-direction clearly showed that the exit flow velocities were perpendicular to the breach section for $Q_r$ = 1 (m$^3$/s). These velocities gave a better view of the separation and contraction zones. In the separation zone that started to form in the main channel before the breach flow exit, the exit velocities in the y-direction were the lowest along the breach section.

The highest values of velocity in the y-direction indicated the zone where flow contraction occurred in the far section of the breach after the separation zone. In general, the distribution of both velocity components along the breach section is almost the opposite, that is, the maximum values of one velocity component correspond to the minimum values of the other component. This is caused by the curvature in the vectors to exit the breach section. Flow tends to exit at the far point of the breach section with a very high velocity in the opposite x-direction for flow entering the breach. This is adjacent to the stagnation point.

The y-velocity decreased with an increase in distance from the flume bottom. Similar distributions were observed at z = 0.04 m, 0.11 m, and 0.20 m. At the far end of the breach in the main channel, y-velocity decreased significantly, similar to x-velocity. Thus, the vectors curved at the exit of the breach section, starting a small distance upstream of the breach end. This caused a change in the distribution of velocities along the channel width.

Both of the velocity components decreased away from the breach and increased near its location. The vector curvature and its impact on the velocity distribution occurred at all levels above the flume bottom. However, results showed that the y-velocities at the breach exit are higher at z = 0.04 m than those at z = 0.11 m. Thus, the curvature of vectors at the breach section was observed to be greater at lower depths of water at the breach section.

For $Q_r = 1$ m³/s and $L_b = 0.25$ m, Figure 5 shows the velocity field for y- and z-velocity components at two locations: x = 0.10 m and x = 0.20 m. Both sections were after the region of the flow separation at the breach, and they showed clearly the impact of the third vertical component of velocity (z-velocity) in the flow pattern near the breach section. This secondary downward current is very strong and obvious at section x = 0.20 m; however, near the end of the breach, its intensity at the other sections in the contraction zone is low. This secondary current caused the vector to drop rapidly, leading to a rapid decrease in the water level in the vicinity of the breach, until the mid-width of the channel, where it started to diminish along the channel width. The z-velocity component decreased with a decrease in z distance at the contraction zone, at x = 0.10 m and 0.20 m.

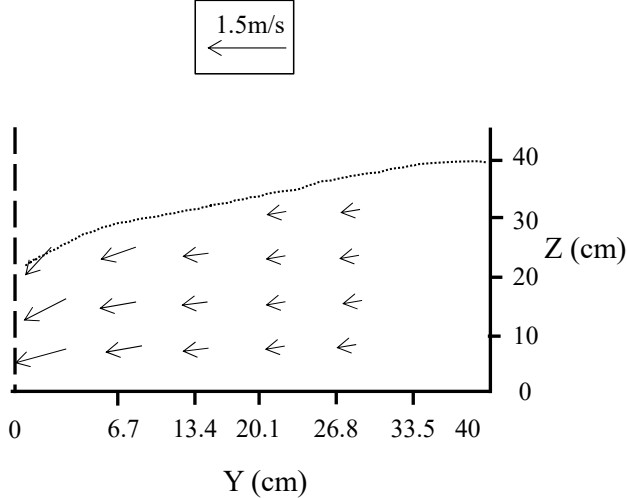

**Figure 5.** y-z velocity vector profile along the breach, $Q_u = 0.16$ m³/s, $Q_r = 1$ m³/s, for $L_b = 0.25$ m, at x = 0.2 m.

### 5.3. Critical Velocity for Sandbags Incipient Motion

Table 3 shows the sandbag dimensions used in the experimental study versus the critical depth averaged velocity required for the onset of sandbag instability. Where non-dimensional critical velocity is defined as $V_{cr}/\sqrt{\frac{(\rho_s - \rho_w)}{\rho_w}gb}$, and $V_{cr}$ is the depth averaged velocity (at mid-section of breach) averaged over the whole flow depth. For spherical sandbags, $a = b = c = D$.

Figure 6 shows the relationship between critical velocity for incipient motion versus the ratio of D/R for the Shields equation, Novak and Nalluri [52] experiments (diameters were from 0.6 mm to 50 mm), and results measured in the present study for spherical sandbags with diameters from 48 to 200 mm. The Shields equation has been plotted using Stricklers' equation for Manning roughness. It can be observed that the critical velocity required for initiation of motion for a single particle is lower than that given by Shields for a given D/R value.

**Table 3.** Sandbag geometric and dimensional properties versus the critical velocity $V_{cr}/\sqrt{\frac{(\rho_s-\rho_w)}{\rho_w}gb}$.

| Experiment | b/c | $\dfrac{V_{cr}}{\sqrt{\frac{(\rho_s-\rho_w)}{\rho_w}gb}}$ | Exp. | b/c | $\dfrac{V_{cr}}{\sqrt{\frac{(\rho_s-\rho_w)}{\rho_w}gb}}$ |
|---|---|---|---|---|---|
| 1 | 0.63 | 0.76 | 27 | 1 | 0.49 |
| 2 | 0.60 | 0.69 | 28 | 1 | 0.51 |
| 3 | 0.50 | 0.69 | 29 | 1 | 0.52 |
| 4 | 0.58 | 0.79 | 30 | 1 | 0.61 |
| 5 | 0.50 | 0.79 | 31 | 1 | 0.52 |
| 6 | 0.42 | 0.93 | 32 | 1 | 0.61 |
| 7 | 0.35 | 0.81 | 33 | 1 | 0.57 |
| 8 | 0.44 | 0.81 | 34 | 1 | 0.53 |
| 9 | 0.44 | 0.81 | 35 | 1 | 0.50 |
| 10 | 0.56 | 0.68 | 36 | 1 | 0.55 |
| 11 | 0.50 | 0.68 | 37 | 1 | 0.53 |
| 12 | 0.45 | 0.72 | 38 | 1 | 0.58 |
| 13 | 0.32 | 1.15 | 39 | 1 | 0.56 |
| 14 | 0.30 | 1.25 | 40 | 1 | 0.55 |
| 15 | 0.23 | 1.25 | 41 | 1 | 0.52 |
| 16 | 0.19 | 1.35 | 42 | 1 | 0.53 |
| 17 | 0.18 | 1.41 | | | |
| 18 | 0.21 | 1.35 | | | |
| 19 | 0.22 | 1.35 | | | |
| 20 | 0.18 | 1.46 | | | |
| 21 | 0.70 | 0.65 | | | |
| 22 | 0.64 | 0.70 | | | |
| 23 | 0.50 | 0.70 | | | |
| 24 | 0.58 | 0.67 | | | |
| 25 | 0.44 | 0.67 | | | |
| 26 | 1.00 | 0.49 | | | |

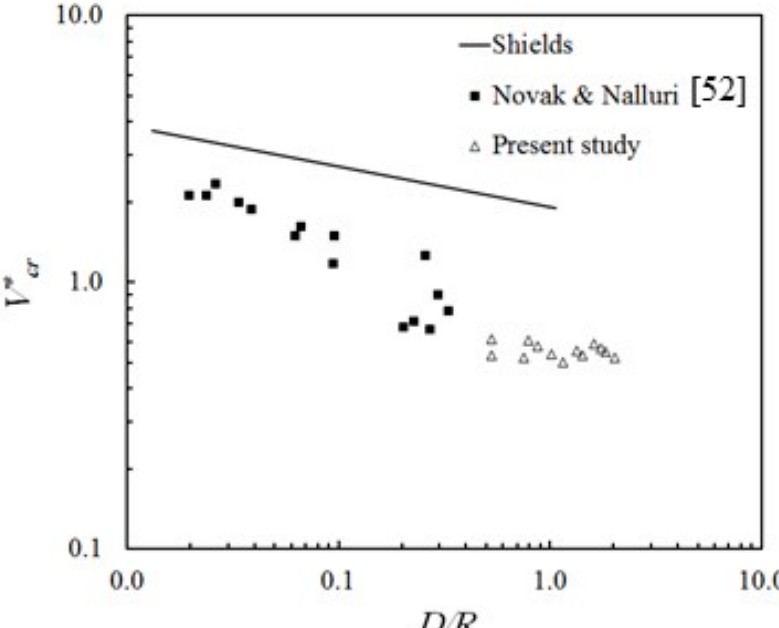

**Figure 6.** Relationship between the non-dimensional critical velocity ($V_{cr}^*$) at the initiation of sandbag motion and *D/R* for circular sandbags.

Experiments in the current study covered a higher range of *D/R* (0.5 < *D/R* < 2) and followed a similar decreasing trend as the Shields equation and experiments of Novak and Nalluri [52]. However, the rate of change of critical velocity with respect to increasing in *D/R* is more pronounced in the

Novak and Nalluri [52] experiments for a $D/R$ lower than 0.3. Using both datasets for critical velocity on spherical particles on smooth and rough beds, and for spherical sandbags, the following relation has been obtained with a coefficient of determination, $R^2$ of 0.90:

$$\frac{V_{cr}}{\sqrt{\frac{(\rho_s - \rho_w)}{\rho_w} gD}} = 0.569 \left(\frac{D}{R}\right)^{-0.34} \tag{7}$$

Introducing the channel hydraulic radius is not common for critical velocity equations describing particle incipient motion, although it has been used by Novak and Nalluri [52] to decrease the effects of flume shape, having used triangular and circular flumes in their experiments. Usually, the main influencing parameters on the critical velocity of a spherical particle are the particle weight and the diameter. The critical velocity measurements for sandbags followed the 1/6th power equation suggested by Stelczer [54], with $R^2$ of 0.95 as shown in Figure 7, where $V_{cr} = 0.381W^{1/6}$ using metric units, and $W$ is the sandbag weight in Newton.

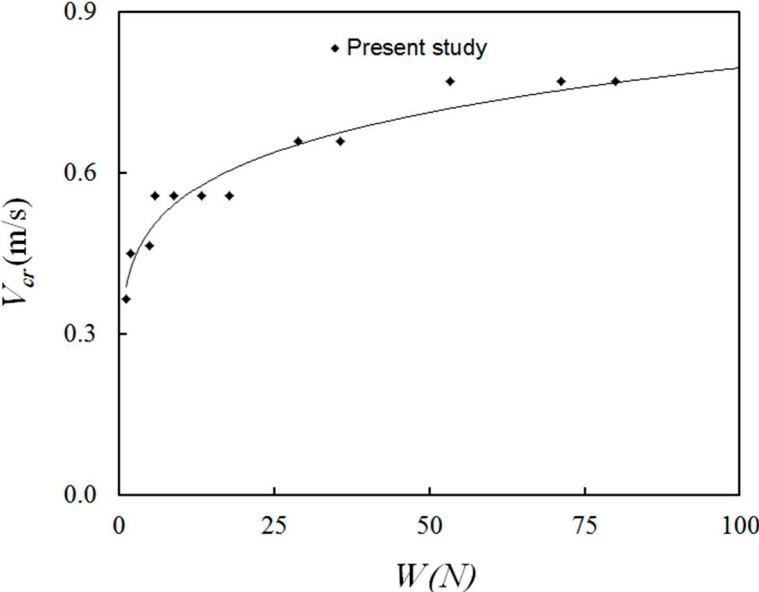

**Figure 7.** Variation of $V_{cr}$ with spherical sandbag weight.

Using the sandbag diameter as a main influencing parameter on its incipient motion has been suggested by numerous studies. Figure 8 shows the results of some of the relevant studies in addition to the experimental findings of the present study on sandbags. It should be noted that the maximum grain diameter used in these studies is 10 mm for the Gulcu [40] and 100 mm for Defne [41]. The critical motion for sandbags followed a very similar trend to those of solitary grains in previous studies, with differences attributed to the conditions and assumptions used in every study (e.g., mixed size grains with equivalent diameter, very small size grains, using big rocks, and rough and smooth beds). Data in Figure 8 suggest that expressions in the form of power function can be used to define the relationship (using metric units) between the sandbag critical velocity and the bag diameter as follows:

$$V_{cr} = 1.633D^{0.47} \tag{8}$$

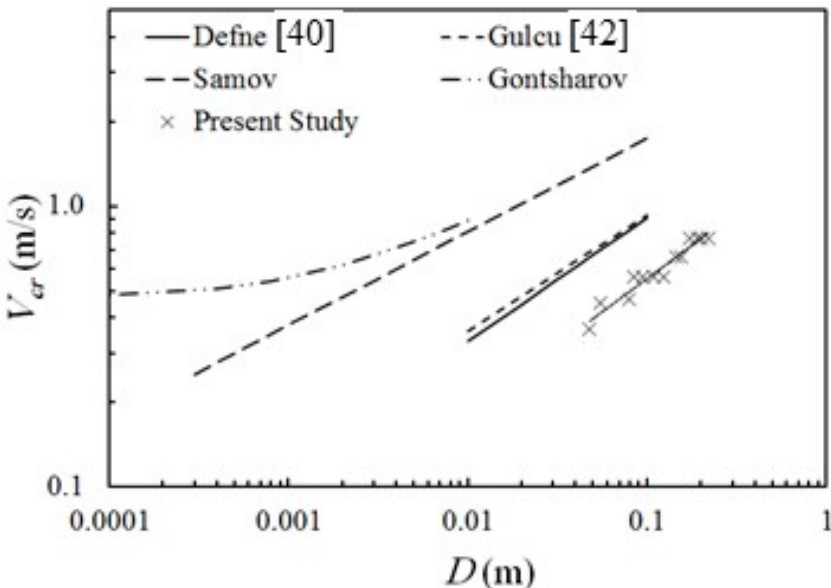

**Figure 8.** Variation of $V_{cr}$ with spherical grain diameter.

Figure 9 shows the relationship between the critical velocity for prism sandbags and the height to length ratio $b/c$, using data from previous studies and this study [41,43].

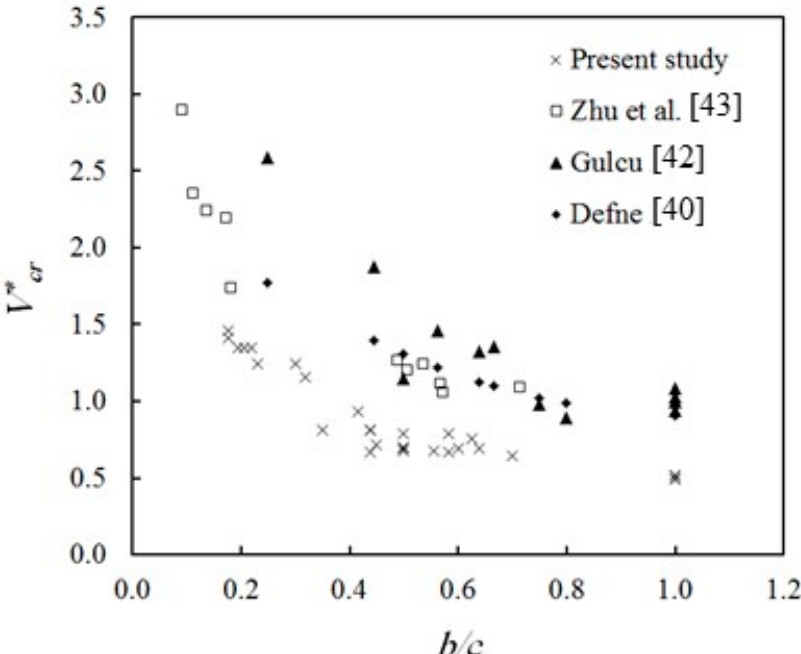

**Figure 9.** $V_{cr}/\sqrt{\frac{(\rho_s-\rho_w)}{\rho_w}gb}$ versus the height to length ratio ($b/c$) for prism sandbags.

The critical velocity required for initiation of particle motion decreases rapidly with the increase of $b/c$ for $b/c \leq 0.7$. For $b/c > 0.7$, the critical velocity remained almost constant until $b/c = 1$ for a cube sandbag. Cube sandbags had the lowest critical velocity and thus were more easily washed away by flowing water than prism sandbags. Gogus and Defne [41] found that the critical velocity required for moving a cube particle is three times lower than for a prism particle, while results of this study found it to be around two times lower. Zhu et al. [43] did not use cube sandbags, and thus no results are reported for $b/c = 1$.

Using the experimental results of Zhu et al. [43] and those measured in the current study, a power function can be used to develop a relationship between critical velocity for sandbag threshold motion and the sandbag height to length ratio with $R^2$ of 0.83 as follows:

$$\frac{V_{cr}}{\sqrt{\frac{(\rho_s - \rho_w)}{\rho_w} g b}} = 0.522 \left(\frac{b}{c}\right)^{-0.67} \tag{9}$$

The database of incipient motion of solitary particles measured in this study and measured previously [43] were used to calibrate some traditional equations and fine tune numerical constants in them by using multiple regression analysis. Table 4 shows these equations with original numeric constants and the same equations after being calibrated with the database.

**Table 4.** Original and calibrated traditional critical velocity equations.

| Study | | Model Equation |
|---|---|---|
| Novak and Nalluri [52] | Original | $V_{cr} = C_3 \left(\frac{D}{R}\right)^{C_4} \sqrt{\frac{(\rho_s - \rho_w)}{\rho_w} g D}$ |
| | Calibrated | $V_{cr} = 0.569 \left(\frac{D}{R}\right)^{-0.34} \sqrt{\frac{(\rho_s - \rho_w)}{\rho_w} g D}$ |
| Brahms | Original | $V_{cr} = C_1 W^{1/6}$ |
| | Calibrated | $V_{cr} = 0.381 W^{1/6}$ |
| Izbash [44] | Original | $V_{cr} = 3.76 \sqrt{D \left(\frac{\rho_s - \rho_w}{\rho_w}\right)}$ |
| | Calibrated | $V_{cr} = 1.77 \sqrt{D \left(\frac{\rho_s - \rho_w}{\rho_w}\right)}$ |

Uncertainty Analysis

In the levee breach closure risk assessment study, the uncertainties of many influencing parameters have to be included. Various choices of sandbag dimensions can influence the outcome in a drastic way, which could lead to the sandbag being swept away by flowing water. Using the calibrated and developed equations for sandbag incipient motion together with the experimental measurements, a quantitative assessment for the uncertainty in the prediction of critical velocity can be presented. The uncertainty analysis defines the prediction error in log cycles as [55–65]

$$e_i = log_{10}(P_i) - log_{10}(T_i) \tag{10}$$

where $e_i$ is the prediction error, $P_i$ is the predicted value of parameter, and $T_i$ is the measured value of the parameter. Data were then used to calculate main indicators defined as mean prediction error $\bar{e} = \sum_{i=1}^{n} e_i$, the width of uncertainty band, $B_{ub} = \pm 1.96 S_e$ and the confidence band around the predicted value:

$$\left\{ P_i \times 10^{\bar{e}+2S_e}, \ P_i \times 10^{\bar{e}-2S_e} \right\} \tag{11}$$

where $S_e$ is the standard deviation of prediction errors and $P_i$ is the predicted value and is taken as unity. Table 5 summarizes the results of the uncertainty analysis performed on both calibrated and developed models for prediction of critical velocity for sandbag motion in flowing water. All models were tested on experimental data in this study and Zhu et al. [43] data.

**Table 5.** Results of prediction uncertainty analysis for sandbag incipient motion models.

| Sandbags Shape | Study | Model | The Width of Uncertainty Band | Prediction Interval Around Hypothetical Predicted $V_{cr}^*$ of Unity |
|---|---|---|---|---|
| Sphere | Novak and Nalluri [52] (Calibrated) | $\dfrac{V_{cr}}{\sqrt{\dfrac{(\rho_s - \rho_w)}{\rho_w} gD}} = 0.569\left(\dfrac{D}{R}\right)^{-0.34}$ | ±0.14 | (0.78–1.51) |
| | Brahms (Calibrated) | $V_{cr} = 0.381 W^{1/6}$ | ±0.05 | (0.91–1.14) |
| | Present study—Sphere | $V_{cr} = 1.633 D^{0.47}$ | ±0.04 | (0.89–1.13) |
| | Samov (Original) | $V_{cr} = 2.5 D^{0.44}$ | ±0.05 | (1.45–1.85) |
| | Izbash [44] (Original) | $V_{cr} = 3.76\sqrt{D\left(\dfrac{\rho_s - \rho_w}{\rho_w}\right)}$ | ±0.05 | (1.87–2.39) |
| | Izbash [44] (Calibrated) | $V_{cr} = 1.77\sqrt{D\left(\dfrac{\rho_s - \rho_w}{\rho_w}\right)}$ | ±0.05 | (0.88–1.12) |
| Prism | Present study—Prism1 | $\dfrac{V_{cr}}{\sqrt{\dfrac{(\rho_s - \rho_w)}{\rho_w} gb}} = 0.295\left(\dfrac{b}{c}\right)^{-1}$ | ±0.19 | (0.52–1.25) |
| | Present study—Prism2 | $\dfrac{V_{cr}}{\sqrt{\dfrac{(\rho_s - \rho_w)}{\rho_w} gb}} = 0.522\left(\dfrac{b}{c}\right)^{-0.67}$ | ±0.08 | (0.91–1.33) |
| | Zhu et al. [43] (Original) | $\dfrac{V_{cr}}{\sqrt{\dfrac{(\rho_s - \rho_w)}{\rho_w} gb}} = 0.44\left(\dfrac{H}{b}\right)^{1/6}\left(\dfrac{b}{c}\right)^{-0.5}$ | ±0.08 | (0.90–1.30) |

The lower the uncertainty in the prediction of a model and the narrower the prediction interval, the more reliable the model is. It is obvious that the widest uncertainty bands for predictions were due to the calibrated [38] proposed equation, with a value of $\pm0.14$ versus $\pm0.05$ for other proposed models for spherical sandbags. The Izbash [44] model, based on spherical grains and rock material, seems to significantly overestimate critical velocity. Other models for spherical sandbags had narrower prediction intervals, with an average of $-10\%$ to $+12\%$ of predicted critical velocity. On the other hand, for prism sandbags, equations seemed to behave in a similar way, with a narrow lower prediction uncertainty of $-10\%$ and slightly wider high prediction uncertainty of $+30\%$. In the following section, it will be shown how the width of prediction intervals can affect the correct choice of the sandbag size.

*5.4. 17th Street Canal Levee Breach Closure*

Hurricane Katrina resulted in a breach of the levees and floodwalls in approximately 20 places. One of the major breaches was on the 17th Street Canal, approximately 305 m from the Old Hammond Highway Bridge, with a breach width of 137 m [5]. The flood protection system in the 17th Street Canal was a concrete floodwall with an I-section over a levee embankment of fill material over a marsh layer [5].

The storm surge moved the levee and floodwall horizontally for about 13.7 m, and this breach—as well as others—accounted for the flooding of the city, the destruction of infrastructure (sewers, water, phone and electricity lines), and the tripping of pumping stations due to water rise. The Army Corps dropped 10 large sandbags to close the breach [29]. They started with 1359 kg (3000 lb) and increased to 2718 kg (6000 lb) using the National Guard Helicopter (see Figure 10). However, the bags were completely washed away, and the initial attempt failed.

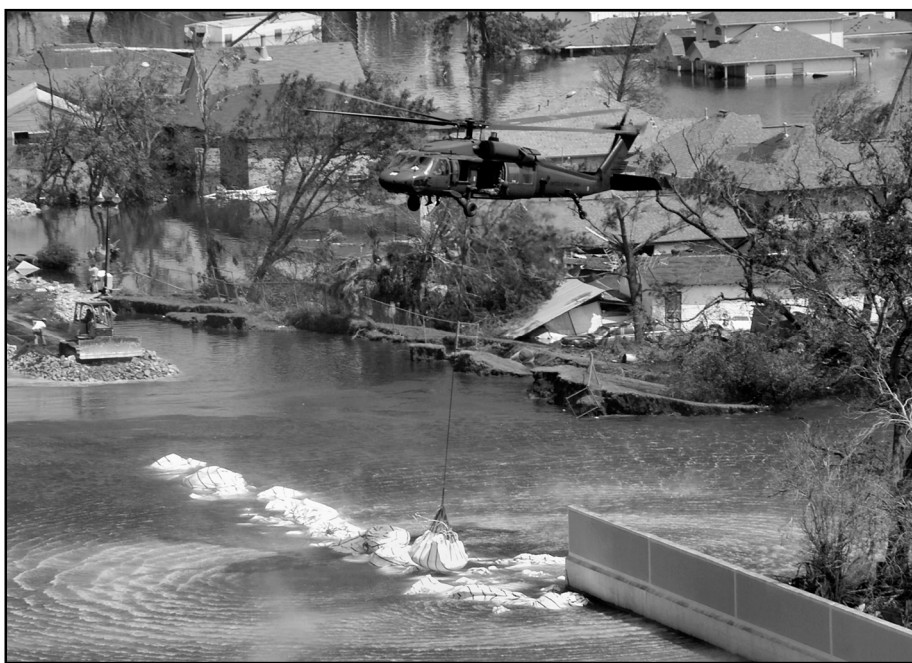

**Figure 10.** Black hawk dumping sandbags to close the 17th Street Canal levee breach [29].

Work in this section is intended to simulate the initial failed attempts of the United States Army Corps of Engineers (USACE) to close the breach using 2718 kg (6000 lb) sandbags on 31 August 2007. While there are several unknowns on the actual failure of the 17th Street Canal floodwalls and levees, Sattar et al. [29] considered a "base scenario" that is more conservative than flow conditions during initial failed attempts.

This scenario uses some of the assumptions used in the base scenario considered by the USACE [5]. The breach is assumed to have widened from 61 to 137 m by 10:00 am. The elevation of water after

the breach had increased by 1.92 m. Corresponding to this water elevation in the channel before the breach, the measured flow across the breach was 580 m$^3$/s. Using these data, the critical velocity is 2.5 m/s (as measured experimentally by Sattar et al. [29] and calculated numerically by Jia et al. [66]) and $V_{cr}$ is 0.78 m/s for 1359 kg (3000 lb) and 0.70 m/s for 2718 kg (6000 lb) sandbags.

Scaling up model parameters to prototype parameters results in significant differences due to model effects, scale effects, and measurement effects [67], with scale effects contributing most to the difference. Making the experimental model as close as possible to prototype dimensions would yield least scale effects. However, it would be uneconomic to build such models. Thus, Heller [67] proposed scales of 1:50 to 1:100 for open channel flows that would be a compromise between both reasonable model size and moderate scale effects. Moreover, Chanson [68] reported that scale effects would be minimized and/or eliminated in models with scales from 1:25 to 1:50. In our experiments, water depth in the flume was 40 cm, which implies a scale of less than 1:50 compared to the 17th Street Canal depth of 7 m. Therefore, it can be assumed that scale effects between the lab setup and case study are minimal and would not affect results.

Calculating the critical velocity from various calibrated and original models for the sandbags that were initially dropped by the USACE [5] during attempts to close the breach showed that some equations provide non-logical results. The critical velocity for sandbag motion as predicted by Samov and Izbash [44] is 2.5–2.8 m/s and 3.1–3.6 m/s for 3000 lb and 6000 lb sandbags, respectively. These velocities greatly exceed the actual critical velocity, suggesting that both sizes of sandbags dropped by the USACE could not have been washed away by flowing water, which is not what happened. Therefore, these equations provided false predictions for sandbag incipient motion and are not recommended to be used without calibration.

The calibrated equations of Brahms and Izbash [44], and the equation developed in the present study, predicted similar values for critical velocity required for the motion of sandbags. Critical velocities had average values from 1.6 m/s to 1.8 m/s for 3000 lb and 6000 lb sandbags, respectively. These predicted velocities were lower than the actual flow velocity when dropping the bags, confirming what really occurred in initial failed attempts where the dropped sandbags could be easily swept away by breach flood waves. These models had the narrowest prediction interval of ±10%, which implies that even when considering the uncertainty in predictions within this interval, the sandbags would still be washed away easily.

As discussed before, spherical and cube shaped sandbags had the lowest critical velocity for initiation of motion among prism sandbags with the same weights. Prism sandbags (with $b/c < 0.4$) have better hydrodynamic characteristics that give them higher stability and resistance to being moved by flowing water. This is confirmed by calculations of critical velocity for prism sandbags that could have been used for initial breach closure attempts.

The actual critical velocity $V_{cr}$ for 1359 kg (3000 lb) and 2718 kg (6000 lb) prism sandbags are 1.10 and 0.90 m/s, respectively. The equations developed in the current study predicted values close to this for critical velocity, which were higher than those predicted by the equation in Zhu et al. [43]. $V_{cr}$ was predicted as 0.83 m/s and 1.04 m/s for developed equations in the present study, compared with 0.37 m/s for Zhu et al. [43]. However, the uncertainty range for the first equation was wider than other equations, where the predicted critical velocity could lie within the range of 0.42–1. On the other hand, the predicted velocity—taking into account the uncertainty range—from the present study was equal to or greater than the actual critical velocity, suggesting that the sandbags would resist being washed away by the flow. The same is true for the 2718 kg (6000 lb) sandbags, where the average critical velocity was higher than or equal to the actual breach flow conditions, and thus, would resist being moved by breach water.

Using this type of prism sandbag, the available helicopters could have been used to carry more sandbags, and breach closure would have been very likely possible in a timely manner. The Zhu et al. [43] model predictions for the critical velocity were lower than those of the fitted model and did not show a clear differentiation between various orientations of sandbag with respect

to flow. The Zhu et al. [43] model's critical velocity was a half less than other models, suggesting a considerable error in sandbag choice.

According to Sattar et al. [29] a sandbag of weight 6795 kg (15,000 lb) was not washed away at about 40% of the breach length and a similar ratio for a sandbag of 3171 kg (7000 lb), while only 20% of the breach required 4530 kg (10,000 lb) sandbag. An experimental simulation [29] and a numerical simulation [66] of the breach flow showed that the depth averaged velocity for the base scenario ranged from 2 to 2.5 m/s. Table 6 presents the critical depth averaged velocity required sandbag motion as predicted by the available models for large spherical sandbags. As discussed above, the Samov equation and Izbash [44] model predicted the highest critical velocity amongst all other models. All models, except for the Novak and Nalluri [52] model, confirmed experimental findings by predicting higher critical velocity for sandbag motion than actual flow conditions; thus, the chosen sandbag was not washed away. This is true when considering the prediction uncertainty interval for each model. Utilizing these results, a recommendation can be made to use the calibrated Brahms model, the model developed in the current study for spherical sandbags, and the calibrated Izbash [44] model to predict the critical velocity for sandbag motion threshold when used in levee breach closure. Considering the three models with a prediction uncertainty of $\pm 10\%$ can give decision makers a rough estimate for the correct choice of sandbag weight and/or diameter for breach closure.

**Table 6.** Critical velocity ($V_{cr}$) for spherical sandbag motion for breach closure in Base Scenario studied by Sattar et al. [29].

| Study | Model | $V_{cr}$ 95% Confidence Interval (m/s) | 10,000 lbs Sandbag | 15,000 lbs Sandbag |
|---|---|---|---|---|
| Novak and Nalluri [52] (Calibrated) | $V_{cr} = 0.569\left(\frac{D}{R}\right)^{-0.34}\sqrt{\frac{(\rho_s-\rho_w)}{\rho_w}gD}$ | Predicted Range | 0.96 (0.73–1.45) | 0.98 (0.75–1.48) |
| Brahms (Calibrated) | $V_{cr} = 0.381W^{1/6}$ | Predicted Range | 2.27 (2.01–2.50) | 2.43 (2.21–2.77) |
| Present study—Sphere | $V_{cr} = 1.633D^{0.47}$ | Predicted Range | 2.00 (1.78–2.26) | 2.16 (1.92–2.44) |
| Samov (Original) | $V_{cr} = 2.5D^{0.44}$ | Predicted Range | 3.00 (4.35–5.55) | 3.24 (4.70–6.00) |
| Izbash [44] (Original) | $V_{cr} = 3.76\sqrt{D\left(\frac{\rho_s-\rho_w}{\rho_w}\right)}$ | Predicted Range | 5.48 (10.25–13.10) | 5.77 (10.79–13.79) |
| Izbash [44] (Calibrated) | $V_{cr} = 1.77\sqrt{D\left(\frac{\rho_s-\rho_w}{\rho_w}\right)}$ | Predicted Range | 2.20 (1.94–2.46) | 2.37 (2.08–2.66) |

## 6. Conclusions

Motivated by the importance of open channel levee breach and the engineering requirements for timely breach closure, this study experimentally investigated some of the hydraulic characteristics of a breach flow together with the effect of sandbag shape and size on the incipient motion of the bags. Experimental measurements during hydraulic modelling in the laboratory showed that prism sandbags have better hydrodynamic characteristics than spherical bags with the same weight. Measurements were used to calibrate existing models and develop new ones for sandbag incipient motion for breach closure. A power function linking critical velocity for sandbag threshold motion and the sandbag height to length ratio was proposed. The developed models for the incipient motion are based on the built experiment that has the following important assumptions: Breach formation and dike failure process are not accounted for, and breach and channel beds are immobile; the breach and channel beds have the same elevation; and the inundation area after a breach is assumed as the open boundary with no effect on breach outflow. Most importantly, similar to previous research, only one sandbag is placed in breach and tested for incipient motion.

Statistical analyses were performed on the prediction models to provide a range of uncertainty in the predicted critical velocity. The available calibrated and developed models were then applied to 17th Street Canal levee breach closure. The obtained results showed that Izbash [44] models predicted the highest critical velocity amongst all other models. All models, except for the Novak and Nalluri [52]

model, confirmed experimental findings by predicting higher critical velocity for sandbag motion than actual flow conditions, and thus, the chosen sandbag was not washed away. Reasonable agreement was achieved in comparing the predicted critical velocity for sandbags with the actual velocities during the USACE attempts for breach closure. Results suggested that some equations can be used considering their uncertainty in prediction to provide a rough view for decision makers on the required sandbags for closing a breach flow. Moreover, through the use of a new formula, it provides engineers with applicable results for implementation in practical cases.

In this work, all experiments for incipient motion of sandbags were carried out in side breach flows to calibrate existing and develop new empirical equations for linking sandbag size and critical flow velocity. Various parameters including breach failure, downstream bed morphological changes, and the associated change in breach flow/velocity, can affect obtained results and could be considered in future research.

**Author Contributions:** A.M.A.S. performed the measurements, drafted the manuscript and designed the figures. H.B. and A.M.A.S. analysed the data. B.G. and A.R.-P. aided in interpreting the results and worked on the manuscript. All authors discussed the results and commented on the manuscript.

**Funding:** This research received no external funding.

**Conflicts of Interest:** The authors declare no conflict of interest.

## Abbreviations

| | |
|---|---|
| $b$ | sandbag height (cm) |
| $B_{ub}$ | the width of the uncertainty band |
| $c$ | sandbag length parallel to the flow direction (cm) |
| $C_{1-8}$ | empirical constants |
| $D$ | the diameter of the spherical particle (mm) |
| $e_i$ | the prediction error |
| $\bar{e}$ | mean prediction error |
| $H$ | water depth above sandbag (m) |
| $L_b$ | breach width (m) |
| $P_i$ | the predicted value of the parameter |
| $Q_b$ | the breach discharge (m$^3$/s) |
| $Q_u$ | main channel discharge (m$^3$/s) |
| $Q_r$ | breach flow ratio = $Q_b/Q_u$ |
| $R$ | channel hydraulic radius (m) |
| $R^2$ | the coefficient of determination |
| $S_e$ | the standard deviation of prediction errors |
| $T_i$ | the measured value of the parameter |
| $V_{cr}^*$ | $V_{cr}/\sqrt{\frac{(\rho_s-\rho_w)}{\rho_w}gb}$; |
| $\rho_s$ | sandbag density (g/cm$^3$) |
| $\rho_w$ | water density (g/cm$^3$) |

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
