# Peer review of "Hydraulic Modeling and Evaluation Equations for the Incipient Motion of Sandbags for Levee Breach Closure Operations"

_water, doi:10.3390/w11020279_

Round 1
Reviewer 1 Report
This manuscript concerns the analysis of laboratory modelling for the evaluation of the empirical equations for the assessment of critical velocity formulae of incipient motion of sandbags for levee breach closure. The experimental dataset is compared with other studies within the same topic. The research herein presented is certainly within the scope of the Journal of Water. However, the manuscript organization, the incompleteness and the lack of clarity/novelty in some parts of the text should be amended prior to publication. My current decision is major revision and I reckon the author to take into consideration all the queries listed below before the manuscript could be considered for publication.
List of comments:
- In the abstract it is said that 3D velocity measurements were performed. However, the results did not show such measurements. Could the authors really show the velocities in the three directions?.
- The introduction is well written however nothing is explained about the morphological collapse of the dikes. Please, take a look at the following reference and expand the explanation about the eroding process and the different numerical strategies for modelling such phenomenon.
A 2D weakly-coupled and efficient numerical model for transient shallow flow and movable bed. C Juez, J Murillo, P García-Navarro. Advances in Water Resources 71, 93-109
- Following with this idea of the geomorphological collapse: the authors have designed the experiments with a steady flow travelling through the lateral breach located in the channel. However, the levee failure is combination of a continuous eroding process and geomorphological collapse, when large chunks of material fall down and the water is flowing as in a dam break. Could the authors elaborate about this issue taken into account their lab setup?.
- Quality of Figure 1 should be improved.
- A table with the hydraulic parameters is missing before the results section.
- In table 4 the authors should justified the two breach widths chosen. How could the authors scaled up these results in reality?.
- Figure 2 should be completely redone. Information given there is really poor. Increase the resolution of the results since you are using an ADV equipment.
- Conclusions section should be re-written. The novelty and real conclusion don’t really arise from those lines.
Author Response
Response to Reviewer One Comments
# | Reviewer One | Authors’ reply |
1 | This manuscript concerns the analysis of laboratory modelling for the evaluation of the empirical equations for the assessment of critical velocity formulae of incipient motion of sandbags for levee breach closure. The experimental dataset is compared with other studies within the same topic. The research herein presented is certainly within the scope of the Journal of Water. However, the manuscript organization, the incompleteness and the lack of clarity/novelty in some parts of the text should be amended prior to publication. My current decision is major revision and I reckon the author to take into consideration all the queries listed below before the manuscript could be considered for publication. | We thank the reviewer for his time and efforts and constructive comments which all have been incorporated in the update manuscript. The authors hope that it will meet the reviewer expectations. |
2 | In the abstract it is said that 3D velocity measurements were performed. However, the results did not show such measurements. Could the authors really show the velocities in the three directions? | Thank you. The reviewer’s comment was addressed accordingly. (Please see Figures 2a and 2b). The authors used ADV for velocity measurements and the machine measures 3D velocity at any given point, in x-, y- and z- directions. The authors have presented velocity in two planar surfaces, i.e., in x-y plane and in y-z plan. This describes the velocity in the three directions. This has been replaced by velocity in x, y and z plans in updated manuscript. We hope this answers the reviewer comment. |
3 | The introduction is well written however nothing is explained about the morphological collapse of the dikes. Please, take a look at the following reference and expand the explanation about the eroding process and the different numerical strategies for modelling such phenomenon. A 2D weakly-coupled and efficient numerical model for transient shallow flow and movable bed. C Juez, J Murillo, P García-Navarro. Advances in Water Resources 71, 93-109 | The authors would like to thank the reviewer for comment. They have considered the proposed reference and cited in manuscript in a brief discussion about the importance of modeling collapse of dikes and dam break flows. However, we would like to point out that in our study, we focused on immobile solid bed, with no eroding material, to investigate the effect of shape and size of sandbags on the threshold of motion, we did not detail morphological collapse of dikes. The main objective of our study was to calibrate existing equation for incipient motion of sandbags and to develop new ones as presented in Tables 3-6. |
4 | Following with this idea of the geomorphological collapse: the authors have designed the experiments with a steady flow travelling through the lateral breach located in the channel. However, the levee failure is combination of a continuous eroding process and geomorphological collapse, when large chunks of material fall down and the water is flowing as in a dam break. Could the authors elaborate about this issue taken into account their lab setup? | We would like to thank the reviewer for this comment. The reviewer is absolutely correct, failure of side protection levees in channels is a gradual process with continuous erosion of bed and widening of levee until a final stable condition is reached, where levee has widened to maximum width and water flow is maximum. This is unlike dam break flow which is a continuous interaction process between erosion; dike morphology and dam break flow. This is confirmed by Hurrican Katrina levees in New Orleans, which all reached to a stable final condition. At this final condition (which is reached shortly after breach), efforts to close the breach with sandbags are attempted. In our work, we are making experiments for incipient motion of sandbags in side breach flows to calibrate existing and develop new empirical equations for linking sandbag size and critical flow velocity. To include breach failure, downstream bed morphological changes, and associated change in breach flow/velocity; this would be an extremely complex situation with various parameters and time would be a factor. A simple model would not be feasible. However this can be considered in future research. |
5 | Quality of Figure 1 should be improved | The requested revision was done accordingly. (Please see Figure 1). |
6 | A table with the hydraulic parameters is missing before the results section. | Thank you for the comment. The reviewer’s comment was responded to accordingly. A table was added in the manuscript. (Please see Table 2). |
7 | In table 4 the authors should justified the two breach widths chosen. How could the authors scaled up these results in reality? | We thank the reviewer for the comment. Two different breach widths are chosen in the experimental setup, one where breach width is around 60% from channel width and the other breach width is 100% from channel width. Authors chose them to represent two possible breaches, a small one and a large one. In many events, e.g., in hurricane Katrina, none of the channel breaches’ width was larger than channel width, they were smaller. This manuscript is an experimental study with measurements done on a small scale prototype, it is not scaled from a real situation. However, its tests situations that are similar to what occurred during this event. Equations produced link the incipient motion of sandbags with breach hydraulics and sandbag hydrodynamic characteristics and have been further tested on real large scale case of 17th street canal breach. |
8 | Figure 2 should be completely redone. Information given there is really poor. Increase the resolution of the results since you are using an ADV equipment. | The requested revision was done accordingly. (Please see Figure 2). |
9 | Conclusions section should be re-written. The novelty and real conclusion don’t really arise from those lines | The requested revision was done accordingly. (Please see Conclusion). |

Reviewer 2 Report
First of all I want to congratulate the authors for such a magnificent work in applied research, always useful to work on such important issues as the effects of floods
I think that the article is well structured and that it meets the requirements of the journal, with a good introduction to the subject, the search of previous cases
I understand that you can not provide more photos but it would have been good a photo of the installation of the experimental levee breach model
A simple drawing would have been good explaining the dimensions, a, b, c, in each case. For example in Table 1
It could also have been accompanied by a small explanatory diagram of the first paragraph of section 5.1., Which describes the protocol followed in the experiments
The experiments were sufficient and yielded a series of results to work with when deciding the Vcr
When is talk about parameters b and c in Figure 5a, I understand that it´s refers to the height of the sandbag and its length, c
In the uncertainty analysis, it is not specified with which tools or programs has been done to obtain the errors
Author Response
Response to Reviewer Two Comments
# | Reviewer Two | Authors’ reply |
1 | First of all I want to congratulate the authors for such a magnificent work in applied research, always useful to work on such important issues as the effects of floods. I think that the article is well structured and that it meets the requirements of the journal, with a good introduction to the subject, the search of previous cases. | We would like to thank the reviewer for his positive opinion about our work and indeed on the time taken to review the work and comments to enhance it. |
2 | A simple drawing would have been good explaining the dimensions, a, b, c, in each case. For example in Table 1 | We thank the reviewer for this comment. The requested revision was done accordingly. We have defined dimensions and in case it is not clear, we would be glad to make an explanatory drawing (Please see Section 3). |
3 | When is talk about parameters b and c in Figure 5a, I understand that it´s refers to the height of the sandbag and its length, c. | |
4 | It could also have been accompanied by a small explanatory diagram of the first paragraph of section 5.1., Which describes the protocol followed in the experiments. The experiments were sufficient and yielded a series of results to work with when deciding the Vcr. | If after updates in manuscript, the reviewer still feels that this figure is important, we will place it in paper. |
5 | In the uncertainty analysis, it is not specified with which tools or programs has been done to obtain the errors | Thank you for the comment. The uncertainty analysis defines the individual prediction error as . The calculated prediction errors for the entire dataset are used to calculate the mean and standard deviation of the prediction errors as and , respectively. A negative mean value indicates that the prediction model underestimated the observed values, and a positive value indicates that the equation overestimated the observed values. Using the values of and Se, a confidence band can be defined around the predicted values of an error using Wilson score method without continuity correction; the use of ±1.96 Se yields an approximately 95% confidence band. These equations are implemented in the excel file and uncertainty bounds were calculated. |

Reviewer 3 Report
The manuscript analyzes the problem of levee breach closure by sandbags with an hydraulic approach considering the prediction of critical velocity for the incipient motion of a sandbag in the breach channel. Experiments are conducted on a physical model in order to study the flow field around the breach channel and to detect the incipient motion velocity of prism and spherical sandbags. A new formulation for the critical velocity is proposed, it is compared with some existing formulae and it is applied to the real case of 17th street canal during Katrina hurricane.
The issues analyzed are of interest either for scientific research and for practical applications. However, my opinion is that the manuscript needs to be improved regarding i) the existing literature review, ii) the experimental tests presentation and results’ discussion, and iii) the application to the real case. In fact, the paper is often not clear in presenting the focus of the study, the experiments presentation and discussion are not complete and refer to findings that are not well supported by the figures, and the application to the real case needs to be more specific and organized in a more logical order. Specific comments are given to the authors.
Finally, the paper may be accepted for publication after a major revision is performed in order to address all the following issues:
- Title is not clear in the aim of “protecting the morphology of the river channel and valley”. Such aims are not dealt in the paper, which is focused on the levee breach closure. I suggest to modify the title as it follows: “Hydraulic modeling and evaluation equations for the incipient motion of sandbags for levee breach closure operations”;
- Lines 38-40: references should be given for levee breach events;
- Line 41: reference should be added to support the similarity between side weir flow and levee breach (e.g., Kamrath et al., 2006 and Michelazzo et al., 2015);
- Lines 113-114: the morphological changes of river channel and valley are not addressed in the paper. For instance, the experimental tests do not deal with movable bed or sediment transport
Section 2: units of measure should be added to every variable
Sections 3-4. Experimental tests should be presented and discussed in a more detailed way:
- the authors should discuss if the physical model is a scale model, the influence of scale effects, the reference to a prototype and how the results scale up to the prototype
- Line 155-156: how much was the Manning coefficient? What was the flume bed material? In which zones of the flume were the 0.01 (unit of measure is missing) water levels measured? For which discharge?
- The same level of breach and main channel may not be the worst configuration for shear stresses in the breach; were other levels taken into account or could the authors provide some comments about this? As a general comment, the authors should explain the choices made for the experimental setup and justify why other configurations were not taken into account (e.g., different bed elevations, roughness of flume and inundation area, movable bed conditions, curved channel, …)
- Provide a clear table explaining the performed tests: how many tests? For which Qu, Qr, Lbr, sandbags dimensions, downstream boundary condition?
- Define the locations where the ADV measures were taken
- Figure 1: should not the coordinate system origin be at the beginning of the breach?
- Table 1: define variables that have not been introduced previously
- in which part of the breach the sandbags were placed?
Section 5. The results’ presentation should be improved and the discussion should refer to the existing literature. In particular, flow zones around breach channels have been analyzed in detailed by Michelazzo et al. (2015 and 2018) for the specific case of zero-height side-weir and breach (that are the same of the present paper).
- provide figures to understand the flow zones in more details and to visualize the water surface around the breach and along the channel
- line 214: to which test are “38 and 42 cm” referred?
- provide explanation for the separation and stagnation zones having the same water level
- line 230: influence of breach width on 3D flow zone have been discussed by Michelazzo et al. (2015 and 2018)
- the 3D flow zones should be detected more precisely and supported by figures and quantitative analyses
- Figure 2a: provide a scale vector velocity (e.g., a 1m/s vector)
- Figures 2a and 2b should be renamed in Figures 2 and 3 respectively, in order to avoid confusion with the subplots a) and b)
- plot water surface in figure 2b
- Figure 2b: “X” or “x”? Use consistent symbols throughout the text
- define Vcr* in figure 3
- is eq.7 defined on dataset of both smooth and rough beds? Can the authors provide justification and implications of deriving an equation on different conditions?
- what’s the x-label of figure 4a?
- Lines 320 and 360: where is USACE (2007) in figure 4b and in table 3?
- Eq. 11: should not the exponent of the lower bound be “-e - 2*Se”?
Section 5.4. As a general comment, the text is not clear when the authors speak about “order of magnitude” in comparing the critical velocity prediction. For instance, at line 405, the difference between the actual velocity (2.5 m/s) and the predicted velocity (1.67 m/s and 1.81 m/s for 3000 lbs sandbag) is not of an order of magnitude but much less. Moreover, the reviewer suggests to organize better this section according to the aim of the paper and to keep only the comparisons that are focused on the aim of the paper. For instance, is table 6 real necessary or could it be deleted? Could tables 4 and 5 be jointed?
- Lines 399 and 409: where are Jia et al (2010) and Stelczer (1981) in table 4?
- Line 425: is “another developed equation” eq. “Present study-1” in table 5? How was this equation derived?
- Line 453: where is Jia et al. (2010) in table 6?
Finally, the authors should clarify how the “calibrated” versions of the eqs in tables 3,4,6 have been derived
Section 6.
Lines 480-483: this sentences are not clear and refer to issued that have not been analyzed in the paper. The reviewer suggests to delete it or to substitute with conclusions more consistent with the analyses performed in the paper.
Add units of measure in the “List of notations”
Reference:
Kamrath, P., Disse, M., Hammer, M., & Koengeter, J. (2006). Assessment of discharge through a dike breach and simulation of flood wave propagation. Natural Hazards, 38(1–2), 63–78. https://doi.org/10.1007/s11069-005-8600-x
Michelazzo, G., Oumeraci, H., & Paris, E. (2015). Laboratory study on 3D flow structures induced by zero-height side weir and implications for 1D modelling. Journal of Hydraulic Engineering, 141(10), 0401502. https://doi.org/10.1061/(ASCE)HY.1943-7900.0001027
Michelazzo, G., Oumeraci, H., & Paris, E. (2018). New hypothesis for the final equilibrium stage of a river levee breach due to overflow. Water Resources Research, 54, 4277–4293. https://doi.org/10.1029/2017WR021378
Author Response
Response to Reviewer Four Comments
# | Reviewer four | Authors’ reply |
1 | The manuscript analyzes the problem of levee breach closure by sandbags with an hydraulic approach considering the prediction of critical velocity for the incipient motion of a sandbag in the breach channel. Experiments are conducted on a physical model in order to study the flow field around the breach channel and to detect the incipient motion velocity of prism and spherical sandbags. A new formulation for the critical velocity is proposed, it is compared with some existing formulae and it is applied to the real case of 17th street canal during Katrina hurricane. The issues analyzed are of interest either for scientific research and for practical applications. However, my opinion is that the manuscript needs to be improved regarding i) the existing literature review, ii) the experimental tests presentation and results’ discussion, and iii) the application to the real case. In fact, the paper is often not clear in presenting the focus of the study, the experiments presentation and discussion are not complete and refer to findings that are not well supported by the figures, and the application to the real case needs to be more specific and organized in a more logical order. Specific comments are given to the authors. Finally, the paper may be accepted for publication after a major revision is performed in order to address all the following issues: | Reviewer four time and effort in reviewing the manuscript is appreciated and comments are all instructive and have been incorporated in the updated manuscript. |
2 | Title is not clear in the aim of “protecting the morphology of the river channel and valley”. Such aims are not dealt in the paper, which is focused on the levee breach closure. I suggest to modify the title as it follows: “Hydraulic modeling and evaluation equations for the incipient motion of sandbags for levee breach closure operations” | Done as per reviewer request. The title is now; Hydraulic modeling and evaluation equations for the incipient motion of sandbags for levee breach closure operations. |
3 | Lines 38-40: references should be given for levee breach events; | Done as per reviewer request. |
4 | Line 41: reference should be added to support the similarity between side weir flow and levee breach (e.g., Kamrath et al., 2006 and Michelazzo et al., 2015) | Done as per reviewer request. |
5 | Lines 113-114: the morphological changes of river channel and valley are not addressed in the paper. For instance, the experimental tests do not deal with movable bed or sediment transport | The reviewer is absolutely correct and we thank him for this comment. We have omitted this part in the updated manuscript. |
6 | Section 2: units of measure should be added to every variable | As per reviewer comment, we have added units beside notations in list of notations. |
7 | Sections 3-4. Experimental tests should be presented and discussed in a more detailed way | We have added some clarifications in the updated manuscript. We hope the reviewer finds the updated description more clearer. |
8 | The authors should discuss if the physical model is a scale model, the influence of scale effects, the reference to a prototype and how the results scale up to the prototype. | We thank reviewer for this comment. The developed model is not a scale model. We have clarified this in the updated manuscript that we are modeling a general case of levee breach and thus there are no scale effects. |
9 | Line 155-156: how much was the Manning coefficient? What was the flume bed material? In which zones of the flume were the 0.01 (unit of measure is missing) water levels measured? For which discharge? | In the updated manuscript, Manning coefficient is 0.01, flume bed and walls are made from immobile material, from rough concrete. |
10 | The same level of breach and main channel may not be the worst configuration for shear stresses in the breach; were other levels taken into account or could the authors provide some comments about this? As a general comment, the authors should explain the choices made for the experimental setup and justify why other configurations were not taken into account (e.g., different bed elevations, roughness of flume and inundation area, movable bed conditions, curved channel, …) | We thank reviewer for this comment. We were attempting to model a general basic case for levee failure and side channel flow and to test motion of sandbags when used for breach closure. Thus, we needed bags to be carried away by moving water and we needed to record the onset of motion, having a breach bed lower than channel would not allow us to capture bag motion as done in same level beds. And sandbag would fall in breach depression. However, this is an important comment and we have added this in conclusion section when discussing model assumptions. |
11 | Provide a clear table explaining the performed tests: how many tests? For which Qu, Qr, Lbr, sandbags dimensions, downstream boundary condition? | Sandbag dimensions are given in Table 1. Number of run tests are given in Table1. For Qr, and Lb, Table 2 is added in the updated manuscript. |
12 | Define the locations where the ADV measures were taken |
If the reviewer wants that the authors place following figure, we would be glad to do so. |
13 | Figure 1: should not the coordinate system origin be at the beginning of the breach? | Done as per requested by reviewer. |
14 | Table 1: define variables that have not been introduced previously | Done as per requested by reviewer. |
15 | In which part of the breach the sandbags were placed? | Sandbags are placed in mid-section of breach. This has been placed in the updated manuscript. Then a sandbag with a selected dimension is placed at the bottom of the flume at the mid-breach section. |
16 | Section 5. The results’ presentation should be improved and the discussion should refer to the existing literature. In particular, flow zones around breach channels have been analyzed in detailed by Michelazzo et al. (2015 and 2018) for the specific case of zero-height side-weir and breach (that are the same of the present paper). | The requested revision was done. |
17 | Provide figures to understand the flow zones in more details and to visualize the water surface around the breach and along the channel | As per the request of the reviewer, we have added Figure 2. We hope that it is clear now and in satisfaction for reviewer. |
18 | Line 214: to which test are “38 and 42 cm” referred? | We thank reviewer for this comment. We have indicated case in the updated manuscript. For Lb=1000px, and Qr=1. |
19 | Provide explanation for the separation and stagnation zones having the same water level | We thank reviewer for this comment, we have added a figure showing the extent of the zones that are formed in open channel side breach. We hope that they are clear for reviewer and readers, especially that we have presented the two studied cases, Qr=1 (downstream channel closed end) and Qr=0.8 (downstream channel opened end). |
20 | line 230: influence of breach width on 3D flow zone have been discussed by Michelazzo et al. (2015 and 2018) - the 3D flow zones should be detected more precisely and supported by figures and quantitative analyses | We thank reviewer for this comment. Proper citation for these two important papers has been added in updated manuscript. |
21 | Figure 2a: provide a scale vector velocity (e.g., a 1m/s vector) | Done as per request of reviewer. |
22 | Figures 2a and 2b should be renamed in Figures 2 and 3 respectively, in order to avoid confusion with the subplots a) and b) | Done as per request of reviewer. |
23 | Plot water surface in figure 2b | Water depth along channel for Lb=1000px, and Qr=1 shown in Figure 2. |
24 | Figure 2b: “X” or “x”? Use consistent symbols throughout the text. | X is used in the manuscript. |
25 | Define Vcr* in figure 3 | As per reviewer request, this item is defined before relevant figure and in list of notations. |
26 | Is eq.7 defined on dataset of both smooth and rough beds? Can the authors provide justification and implications of deriving an equation on different conditions? | This equation has been described by the literature of Novak and Nalluri and not by the authors. |
27 | What’s the x-label of figure 4a? | We thank the reviewer for this comment |
28 | Lines 320 and 360: where is USACE (2007) in figure 4b and in table 3? | The figure shows a picture for attempts by the USACE to close breach. The table contains the results of critical velocity using the real sandbag weights used by the ASCE. |
29 | Eq. 11: should not the exponent of the lower bound be “-e - 2*Se”? | We thank the reviewer for this comment. We have corrected this in the updated manuscript. |
30 | Section 5.4. As a general comment, the text is not clear when the authors speak about “order of magnitude” in comparing the critical velocity prediction. For instance, at line 405, the difference between the actual velocity (2.5 m/s) and the predicted velocity (1.67 m/s and 1.81 m/s for 3000 lbs sandbag) is not of an order of magnitude but much less. Moreover, the reviewer suggests to organize better this section according to the aim of the paper and to keep only the comparisons that are focused on the aim of the paper. For instance, is table 6 real necessary or could it be deleted? Could tables 4 and 5 be jointed? | As per request of reviewer, we have removed the term ‘order of magnitude’ from the updated manuscript. |
31 | Line 425: is “another developed equation” eq. “Present study-1” in table 5? How was this equation derived? | This equation was developed using the experimental results of Zhu et al. [43] and the present study experimental measurements between critical velocity for sandbag threshold motion and the sandbag height to length ratio with R2 of 0.83. This is in the manuscript in description of equation 9. |
32 | Finally, the authors should clarify how the “calibrated” versions of the eqs in tables 3,4,6 have been derived | This has been done by simple multiple regression. |
33 | Section 6. Lines 480-483: this sentences are not clear and refer to issued that have not been analyzed in the paper. The reviewer suggests to delete it or to substitute with conclusions more consistent with the analyses performed in the paper. | Done as per request of reviewer. |
34 | Add units of measure in the “List of notations” | Done as per request of reviewer. |
35 | Reference: Kamrath, P., Disse, M., Hammer, M., & Koengeter, J. (2006). Assessment of discharge through a dike breach and simulation of flood wave propagation. Natural Hazards, 38(1–2), 63–78. https://doi.org/10.1007/s11069-005-8600-x
Michelazzo, G., Oumeraci, H., & Paris, E. (2015). Laboratory study on 3D flow structures induced by zero-height side weir and implications for 1D modelling. Journal of Hydraulic Engineering, 141(10), 0401502. https://doi.org/10.1061/(ASCE)HY.1943-7900.0001027
Michelazzo, G., Oumeraci, H., & Paris, E. (2018). New hypothesis for the final equilibrium stage of a river levee breach due to overflow. Water Resources Research, 54, 4277–4293. https://doi.org/10.1029/2017WR021378 |

Reviewer 4 Report
The reviewer wants to thank the authors for their paper about the motion of sandbags, which they have experimentally investigated. He/she has some suggestions/comments/questions, which should be addressed by the authors:
- The citation style is not according the regulation of the water. Please correct this in the paper as well as in the References.
- Line (l)135-154: Are the test really conducted at two Universities with the same experimental set-up. This should be clarified.
- L156: discharge measured by only one manometer? It shows the differential pressure. But why is in the Fig.1 one electromagnetic flow meter presented. Why is this one not used?
- L158: include this 3 m in the Figure 1, so that the location is clear.
- L162: Q_{in} is fixed with 0.16 *2 m3/s and the downstream gates regulate the splitting in connection with the downstream height. Please give the minimum and maximum water depth in the channel behind the breach as well as a relation between the two values (water height and discharge remaining) to have a feeling for the flow speed behind the breach. … more information than given in line 2010-217 and also for both breach width.
- Please also include the Froude numbers before and the range after the breach.
- How far is the gate downstream of the breach? Does it have an influence of the flow in the breach because of a backflow in a high position? How height was the max position in relation to the remaining water depth over the gate?
- Fig.1 the presented measurements should concentrate on the experimental set-up and not only the outsides.
- L164: the 60 mm honeycomb is a good solution to reduce secondary flow speeds but seem to be not very long. But it doesn’t help in case of the incorrect distribution of the main flow. Was the flow after the honeycomb checked as well as upstream of the breach so that a fully developed velocity profile?
- Fig.1 the printed coordinate system doesn’t fit the description in the text. If the reviewer is correct, the positive z-axis would point downwards. Please show it in the correct position of the origin.
- How does the water flow back into the storage after it flows out of the breach? Is there an additional step indicated by the rectangle with the arrows? How is this outflow boundary condition checked and modelled?
- Fig.1 a photo of the actual set-up would help.
- L187 and ongoing: discharge is measured incoming (it is assumed that two devices are used) and at least one more measurement is needed. Is the discharge measured over the slice gate on the downstream side?
- L191: this initial one is specified by which condition?
- How many bags are inserted? A closed line or only one? How is motion defined in this relation? Is it enough, when it changes the shape or is a small movement enough?
- Why is it reduced in three steps?
- Does it make a difference if the filled bags are filled with water or not? How did the authors consider this?
- L198: the bag(s) is/are inserted again and the critical conditions are reach again to measure the velocity profile. Where was this conducted in the breach?
- General: the reviewer doesn’t understand why this analysis has to be conducted in this complex situation of the breach. It could be easily done in the flume upstream with only longitudinal velocities and then tested in the breach, where there is a 3D-veloiticy distribution to check the conducted investigation.
- L188 as breach width 0.25 and 0.4 m are used. … please check the whole paper that all given values are correctly associated with the correct width and given for both.
- Figure 2a and 2 b should be 2 and 3. The axis are not clear and furthermore those cuts should be indicated in Fig. 1 … the reviewer will check these figures after this was clarified.
- L236 vectors and no streamlines. Please provide a reference vector with a defined length in the Figures.
- Sect 5.2 why is the full discharge through the breach case chosen to be shown?
- Figure 3 wrong citation style and only include the circular bags? Please clarify this also in the caption of the Figure.
- Figure 3 and 4b please don’t use logarithmic axis.
- General a 22 cm sphere should fill nearly half of the breach width … are those sandbags fully filled so that they remain their shape? In this case wouldn’t they roll way? So a certain deformation should be considered but how does the authors include this?
- Figure 5 b: did the authors capture this picture? Source?
- Please include in the discussion a section with the model assumption.
The influence of the shape change of the sandbag in the experiment should be addressed as well as the missing information about the experimental set-up. If those points are cleared, the reviewer is happy to read the paper again and will give further comments to the results as well as the presented case study.
Author Response
Response to Reviewer Three Comments
# | Reviewer Three | Authors’ reply |
1 | The reviewer wants to thank the authors for their paper about the motion of sandbags, which they have experimentally investigated. He/she has some suggestions/comments/questions, which should be addressed by the authors: | The authors thank the reviewer for his effort in reviewing the manuscript and detailed comments, which are incorporated in the updated manuscript. |
2 | The citation style is not according the regulation of the water. Please correct this in the paper as well as in the References. | Done as per the request of the reviewer. |
3 | Line (l)135-154: Are the test really conducted at two Universities with the same experimental set-up. This should be clarified. | We would like to thank the reviewer for this comment. Initial runs were made at the University of South Carolina, but the actual experimental setup and measurements were made at Cairo University. In order to avoid confusion, we have removed the part of South Carolina and updated this part. (Please see Section 3). |
4 | L156: discharge measured by only one manometer? It shows the differential pressure. But why is in the Fig.1 one electromagnetic flow meter presented. Why is this one not used? | We would like to thank the reviewer for this comment. The reviewer is absolutely correct, the authors have updated the manuscript so that it is clear that the discharge is measured by the electromagnetic flow meter installed on discharge pipe. (Please see Section 3). |
5 | L158: include this 3 m in the Figure 1, so that the location is clear. | Done as per the request of the reviewer. |
6 | Please give the minimum and maximum water depth in the channel behind the breach as well as a relation between the two values (water height and discharge remaining) to have a feeling for the flow speed behind the breach. … more information than given in line 210-217 and also for both breach width. | We thank reviewer for this comment. We have added following figure for clarification.
|
7 | Please also include the Froude numbers before and the range after the breach. | Done as per the request of the reviewer. |
8 | How far is the gate downstream of the breach? Does it have an influence of the flow in the breach because of a backflow in a high position? How height was the max position in relation to the remaining water depth over the gate? | The authors thank the reviewer for this comment. The authors have updated figure 1 as per request of reviewer to be clearer, the downstream gate is at 3m from breach location. When all channel flow is directed through breach (80% from channel flow), the water is stagnant and a recirculation zone is developed at the downstream as shown in below figure. There is no effect on outlet boundary on breach flow as it is an open boundary. |
9 | Fig.1 the presented measurements should concentrate on the experimental set-up and not only the outsides. | Done as per the request of the reviewer. |
10 | L164: the 60 mm honeycomb is a good solution to reduce secondary flow speeds but seem to be not very long. But it doesn’t help in case of the incorrect distribution of the main flow. Was the flow after the honeycomb checked as well as upstream of the breach so that a fully developed velocity profile? | The authors thank the reviewer for this comment. Yes, the authors have made checks on velocity profile and taken velocity measurements before breach by 500px and by 1m and found a satisfying developed vertical flow profile. |
11 | Fig.1 the printed coordinate system doesn’t fit the description in the text. If the reviewer is correct, the positive z-axis would point downwards. Please show it in the correct position of the origin. | Done as per request of reviewer. |
12 | How does the water flow back into the storage after it flows out of the breach? Is there an additional step indicated by the rectangle with the arrows? How is this outflow boundary condition checked and modelled? | Ans. Thank you for the comment. The outflow is a free boundary where water flowing out of the breach is allowed to flow freely without steps or any obstacles. It does not affect the water out of the breach by any means. There are no additional steps in water way when it exits breach; it is allowed to leave freely. Thus, there is no need to model any outflow boundary.
|
13 | Fig.1 a photo of the actual set-up would help. | Ans. Thank you for the comment. The quality of figure 1 was improved. |
14 | L187 and ongoing: discharge is measured incoming (it is assumed that two devices are used) and at least one more measurement is needed. Is the discharge measured over the slice gate on the downstream side? | We thank the reviewer for this comment. The discharge is measured at the upstream using the electromagnetic flow meter, this gives us the total flow entering the channel. When all flow is allowed through breach, no other measurement is needed. When only a portion of flow directed to breach, a downstream gate is used to do this job. Gate is calibrated through independent experiments to determine flow throw it. |
15 | L191: this initial one is specified by which condition? | We thank the reviewer for this comment. The initial conditions prior to each test are described in manuscript by steady conditions, when the flow is stabilized in the whole experimental setup. This means when flow is developed before the breach and when several repeated experimental measurements at same location yield same result for water depth and velocities. |
How many bags are inserted? A closed line or only one? How is motion defined in this relation? Is it enough, when it changes the shape or is a small movement enough? | In our experiments, we used single solitary sandbag with pre-set shapes. Only one was tested at a time for incipient motion. Motion is defined when bag starts to move under the effect of flow shear stress. The sandbag does not change in shape, it remains with the pre-set shape during the whole experiment. | |
Why is it reduced in three steps?
| In order to define the incipient motion threshold, which is the velocity at which sandbag starts to move. Authors chose a procedure mentioned in literature, which is adopted by Gogus and Defne (2005) and Gulcu (2009). We have added this in updated manuscript. Gogus M, Defne Z. Effect of shape on incipient motion of large solitary particles. Journal of Hydraulic Engineering. 2005; 131 (1), 38–45. Gulcu B. Incipient motion of coarse solitary particles, MSc. Thesis, The Graduate School of Natural and Applied Sciences of Middle East Technical University, Ankara, Turkey, 2009. | |
Does it make a difference if the filled bags are filled with water or not? How did the authors consider this? | Bags were made from cloth similar to that used in real sandbags, in both when bag is placed in water, water shall penetrate bag to the sand inside. This automatically occurs in the experiments. So the experiments imitate real case. | |
L198: the bag(s) is/are inserted again and the critical conditions are reach again to measure the velocity profile. Where was this conducted in the breach? | Bag is placed in mid-section of breach. This is placed in the updated manuscript. | |
General: the reviewer doesn’t understand why this analysis has to be conducted in this complex situation of the breach. It could be easily done in the flume upstream with only longitudinal velocities and then tested in the breach, where there is a 3D-veloiticy distribution to check the conducted investigation. | We thank reviewer for this comment. We wanted to model the same case and in previous work by the authors, it was found that the distribution of velocity in breach flow (weir side flow) is different from that in developed open channel flow as shown in below figure. | |
L188 as breach width 0.25 and 0.4 m are used. … please check the whole paper that all given values are correctly associated with the correct width and given for both. | Done as per the request of the reviewer. | |
Figure 2a and 2 b should be 2 and 3. The axis are not clear and furthermore those cuts should be indicated in Fig. 1 … the reviewer will check these figures after this was clarified. | Done as per the request of the reviewer. We hope that they are satisfactory in the updated manuscript. | |
L236 vectors and no streamlines. Please provide a reference vector with a defined length in the Figures. | Done as per the request of the reviewer. Streamlines are replaced by vectors in the updated manuscript. A reference vector is added in figure. | |
Sect 5.2 why is the full discharge through the breach case chosen to be shown? | We chose results of this case to be presented since they give higher flow velocity through the breach and thus plots of velocity vectors would show more clearer the direction of flow in 3D through the breach. | |
Figure 3 wrong citation style and only include the circular bags? Please clarify this also in the caption of the Figure. Figure 3 and 4b please don’t use logarithmic axis. | As per reviewer comment, the authors have updated the figure caption to show that it includes circular sandbags. Regarding changing the figures axes to be normal instead of logarithmic, this would not be good in figure presentation. | |
General a 22 cm sphere should fill nearly half of the breach width … are those sandbags fully filled so that they remain their shape? In this case wouldn’t they roll way? So a certain deformation should be considered but how does the authors include this? | We would like to thank the reviewer for this comment. Sandbags with different shapes are fully filled with sand prior to being woven so that shape of sandbag does not deform. No deformation was observed during experiments. | |
Figure 5 b: did the authors capture this picture? Source? | This figure is from Sattar et al. (2008). This has been placed in the updated manuscript. Sattar AMA, Kassem A, Chaudhry MH. Case study: 17th street canal breach closure procedures. Journal of Hydraulic Engineering. 2008; 134 (11), 1547–1558. | |
Please include in the discussion a section with the model assumption. | Done as per requested by the reviewer in the conclusion section. | |
The influence of the shape change of the sandbag in the experiment should be addressed as well as the missing information about the experimental set-up. | We would like to thank the reviewer for this comment. Sandbags with different shapes are fully filled with sand prior to being woven so that shape of sandbag does not deform. No deformation was observed during experiments. |
Round 2
Reviewer 1 Report
All my comments have been taken into consideration.
Author Response
We thank the reviewer for his time and efforts.

Reviewer 3 Report
The authors replied most of the comments raised by the reviewer. The manuscript has been improved and the analyses developed are now described and presented better. However, some issues have not been addressed fully.
Moreover, there are some problems with the files provided by the authors. In particular, the new version of the manuscript (water-406055-peer-review-v2.pdf) does not include some of the responses to some issues raised by the reviewer, even if the authors declare that they’ve addressed the comments in the response-to-reviewer file. Further, the supplementary file “water-406055-supplementary.docx” is different from the revised manuscript and, therefore, it is not clear its connection with the manuscript.
Therefore, I recommend a minor revision before the paper publication.
In particular, the authors are invited to carefully address the following issues:
- it is not clear which references for flood events of Section 1 have been added and if reference [1] includes all the levee breach events cited by the authors
- no specific comments about scale effects can be found in the revised version of the manuscript. However, even if the model is not a scale model, it is evident that the laboratory scale is quite different from the real case scale and that scale effects could arise. Since the results are used for real cases prediction, the authors should give some comments to support the applicability of their experimental findings to the real scale
- the useful comments made by the authors about the experiments’ assumptions should be added also in the Section that introduces the experimental setup and not only in the Conclusions
- why Frd was not defined in Table 2 when Qr=1? How can Fru be constant regardless variations of breach width and Qr?
- the figure proposed by the authors to locate ADV measurements is useful, and the reviewer agrees to add it to the revised paper. Anyway, a reference to the locations depicted in the proposed figure should be added in the text. Moreover, the following corrections should be taken into account:
i) add “X=0.25”
ii) place the coordinate system exactly where X=0.0 and Y=0.0
iii) remove the background squared gray grid
- the above comments about the placing of coordinate system and the removing of the background squared gray grid should be applied to the other similar figures
- the authors should comment further the choice of the sandbag location, since the mid-section could not be that one with the maximum outflow velocity
- reference Michelazzo et al. (2018) should be added to the References Section. “Michelazzo, G., Oumeraci, H., & Paris, E. (2018). New hypothesis for the final equilibrium stage of a river levee breach due to overflow. Water Resources Research, 54, 4277–4293. https://doi.org/10.1029/2017WR021378”
- regarding the figures’ naming, please use letters “a”, “b”, etc only in case of subplots of the same figure; otherwise, use progressive numbers
- figure 2 is a good representation of water surface variation. However, the reviewer intended to plot water surface in figure showing y-z velocity profiles (figure 4b in the revised manuscript) in order to visualize the flow toward the breach
- it is still not clear where USACE model is reported when predicting the critical velocity. For instance, it should be in table 4 according to line 435, but it can’t be found. In the same way, where can the Stelczer velocity, that is cited at line 487, be found in table 5? The authors are invited to solve this issue or to explain it better
- no replies are given by the authors to reviewer’s comments about organization of Section 5.4 and presentation of tables 4-5-6
- the calibration methodology of the equations should be explained in the manuscript
- references should be checked in order to be consistent with the citations made in the main text. Moreover, the same style for references should be used throughout the paper
- check language at lines 20-21, 67, 113-114, 115-116, 533. In general, check for typos and grammatical errors
Author Response
1 | The authors replied most of the comments raised by the reviewer. The manuscript has been improved and the analyses developed are now described and presented better. However, some issues have not been addressed fully. | We would like to thank the reviewer for time and efforts that improved the manuscript. |
2 | Moreover, there are some problems with the files provided by the authors. In particular, the new version of the manuscript (water-406055-peer-review-v2.pdf) does not include some of the responses to some issues raised by the reviewer, even if the authors declare that they’ve addressed the comments in the response-to-reviewer file. Further, the supplementary file “water-406055-supplementary.docx” is different from the revised manuscript and, therefore, it is not clear its connection with the manuscript. Therefore, I recommend a minor revision before the paper publication. | Updated revision will eliminate this problem. |
3 | - it is not clear which references for flood events of Section 1 have been added and if reference [1] includes all the levee breach events cited by the authors | This has been updated in the revised manuscript. |
4 | - no specific comments about scale effects can be found in the revised version of the manuscript. However, even if the model is not a scale model, it is evident that the laboratory scale is quite different from the real case scale and that scale effects could arise. Since the results are used for real cases prediction, the authors should give some comments to support the applicability of their experimental findings to the real scale | We have added a discussion about scale effects in updated manuscript as per request of reviewer. |
5 | - the useful comments made by the authors about the experiments’ assumptions should be added also in the Section that introduces the experimental setup and not only in the Conclusions | Done as per request of reviewer |
6 | - why Frd was not defined in Table 2 when Qr=1? How can Fru be constant regardless variations of breach width and Qr? | The authors thank the reviewer for this comment. In case of Qr=1, all upstream channel flow is directed towards breach and the downstream end of the channel after the breach has stagnant water and thus no Fr. The upstream flow to the experiment is kept constant in all breach widths and this gives similar velocity, thus Fr does not change. |
7 | - the figure proposed by the authors to locate ADV measurements is useful, and the reviewer agrees to add it to the revised paper. Anyway, a reference to the locations depicted in the proposed figure should be added in the text. Moreover, the following corrections should be taken into account: i) add “X=0.25” ii) place the coordinate system exactly where X=0.0 and Y=0.0 | Done as per request of reviewer |
8 | - the above comments about the placing of coordinate system and the removing of the background squared gray grid should be applied to the other similar figures | Done as per request of reviewer |
9 | - the authors should comment further the choice of the sandbag location, since the mid-section could not be that one with the maximum outflow velocity | We thank the reviewer for this comment. Based on previous experience with work in breach closure with sandbags, Sattar (2009) extensively studied various procedures for closure of levee breaches and found that staring by dumping sandbags in mid breach sections would be the most effective way for closure. This has been clarified in the revised manuscript. |
10 | - reference Michelazzo et al. (2018) should be added to the References Section. “Michelazzo, G., Oumeraci, H., & Paris, E. (2018). New hypothesis for the final equilibrium stage of a river levee breach due to overflow. Water Resources Research, 54, 4277–4293. https://doi.org/10.1029/2017WR021378” | Done as per request of reviewer |
11 | - regarding the figures’ naming, please use letters “a”, “b”, etc only in case of subplots of the same figure; otherwise, use progressive numbers | Done as per request of reviewer |
12 | - figure 2 is a good representation of water surface variation. However, the reviewer intended to plot water surface in figure showing y-z velocity profiles (figure 4b in the revised manuscript) in order to visualize the flow toward the breach | Done as per request of reviewer |
13 | - it is still not clear where USACE model is reported when predicting the critical velocity. For instance, it should be in table 4 according to line 435, but it can’t be found. | The authors thank reviewer for this comment. In the updated manuscript, this has been corrected to Gulcu (2009). |
In the same way, where can the Stelczer velocity, that is cited at line 487, be found in table 5? The authors are invited to solve this issue or to explain it better | The authors thank reviewer for this comment. In the updated manuscript, this has been corrected to Novak and Nalluri 1984. | |
14 | - no replies are given by the authors to reviewer’s comments about organization of Section 5.4 and presentation of tables 4-5-6 Section 5.4. As a general comment, the text is not clear when the authors speak about “order of magnitude” in comparing the critical velocity prediction. For instance, at line 405, the difference between the actual velocity (2.5 m/s) and the predicted velocity (1.67 m/s and 1.81 m/s for 3000 lbs sandbag) is not of an order of magnitude but much less. Moreover, the reviewer suggests to organize better this section according to the aim of the paper and to keep only the comparisons that are focused on the aim of the paper. | The authors apologize for not addressing this comment. Done as per request of reviewer. Discussion updated and many parts are omitted in the updated manuscript. |
15 | For instance, is table 6 real necessary or could it be deleted? Could tables 4 and 5 be jointed? | The authors apologize for not addressing this comment. In the updated manuscript, tables 4 and 5 are deleted and information about critical velocities has been placed in text. |
16 | - the calibration methodology of the equations should be explained in the manuscript | Done as per request of reviewer |
17 | - references should be checked in order to be consistent with the citations made in the main text. Moreover, the same style for references should be used throughout the paper | Done as per request of reviewer |
18 | - check language at lines 20-21, 67, 113-114, 115-116, 533. In general, check for typos and grammatical errors | Done as per request of reviewer |

Reviewer 4 Report
The reviewer wants to thank the authors for their corrections.
- The coordinate system in the figure is not correct described. Please correct this. Line 172 x in positive main direction ok but y point AWAY from the side with the breach and currently the z points downwards. Please check this very very carefully and look up Right -hand rule.
- Figure 1 delete the gate direct after the honey comb. This one is not used and only makes it confusing + add a cutting through the section of the breach, so that it is clear how the ground is model in the breach case.
- The photo is missing and was not added. Why? It would be very easy and would help to understand the set-up.
- The reviewer agrees with the authors, that the flow in the breach is more complex than in a straight direction and that is exactly the reason, why he/she would have suggested, that the investigations are first conducted in a simple flow and in a second step reproduced in the breach. This would reduce the uncertainties of the investigation significantly. Please add this at least in the discussion part and argue why it was done in this way.
- The reviewer questions this “sudden” filling of the bags. There are a lot of assumptions made and those have to be discussed in a separate section and not only in two sentence in the conclusion.
- There should not be a figure 4a (a)!
- The current Fig 4a und 4b should show the similar design and axis. Please also make walls visible.
- Reference vector in 4a is missing.
- 22 cm ball and no deformation? If this is a perfect ball, why should this bag stand still for only one second and not roll away. The reviewer should think carefully about the formulation of the answers and clarify this also in the text.
- Do the authors have the permission of the original author of the picture in Figure 7b?
The reviewer is looking forward to read the corrected version.
Author Response
1 | The reviewer wants to thank the authors for their corrections. | The authors thank the reviewer for his time given for this manuscript with constructive comments that helped enhancing quality of work. |
2 | - The coordinate system in the figure is not correct described. Please correct this. Line 172 x in positive main direction ok but y point AWAY from the side with the breach and currently the z points downwards. Please check this very very carefully and look up Right -hand rule. | Done as per request of reviewer |
3 | - Figure 1 delete the gate direct after the honey comb. This one is not used and only makes it confusing + add a cutting through the section of the breach, so that it is clear how the ground is model in the breach case. | Done as per request of reviewer |
4 | - The photo is missing and was not added. Why? It would be very easy and would help to understand the set-up. | Done as per request of reviewer |
5 | - The reviewer agrees with the authors, that the flow in the breach is more complex than in a straight direction and that is exactly the reason, why he/she would have suggested, that the investigations are first conducted in a simple flow and in a second step reproduced in the breach. This would reduce the uncertainties of the investigation significantly.
Please add this at least in the discussion part and argue why it was done in this way. | Done as per request of reviewer |
6 | There are a lot of assumptions made and those have to be discussed in a separate section and not only in two sentence in the conclusion. | We thank reviewer for this comment. We have added assumptions in experiment description. |
7 | - There should not be a figure 4a (a)! | Done as per request of reviewer |
8 | - The current Fig 4a und 4b should show the similar design and axis. Please also make walls visible. | Done as per request of reviewer |
9 | - Reference vector in 4a is missing. | Done as per request of reviewer |
10 | - 22 cm ball and no deformation? If this is a perfect ball, why should this bag stand still for only one second and not roll away. The reviewer should think carefully about the formulation of the answers and clarify this also in the text. | The work in this study deals with sandbags, rather than perfect geometric prisms, i.e. the circular sandbag is not a perfect sphere/ball; or else, it would not imitate the actual sandbag used in real sites. This work followed procedure performed by Zu et al (2004) and Gulcu (2009), Gogus and Defne (2005), and Defne (2002) by defining the geometry of a sandbag. Where a, b, and c are dimensions of a prism enfolding sandbag. This has been added to the updated manuscript. |
11 | - Do the authors have the permission of the original author of the picture in Figure 7b? | We thank reviewer for this comment, this picture is referenced/ originally used by the first author in Sattar (2009) paper and cited here with permissions from the journal. |

Round 3
Reviewer 4 Report
The reviewer wants to thank the authors for their corrections in the paper. The picture of the complete experimental set-up is missing again but he/she assumes that the authors don’t want to publish such a picture. OK. Thank you for the further corrections.